# Microcavity phonoritons – a coherent optical-to-microwave interface

Alexander Sergeevich Kuznetsov [1] ✉, Klaus Biermann[1], Andres Alejandro Reynoso[2,3,4], Alejandro Fainstein [2,3] & Paulo Ventura Santos [1]

Optomechanical systems provide a pathway for the bidirectional optical-to-microwave interconversion in (quantum) networks. These systems can be implemented using hybrid platforms, which efficiently couple optical photons and microwaves via intermediate agents, e.g. phonons. Semiconductor exciton-polariton microcavities operating in the strong light-matter coupling regime offer enhanced coupling of near-infrared photons to GHz phonons via excitons. Furthermore, a new coherent phonon-exciton-photon quasiparticle termed phonoriton, has been theoretically predicted to emerge in microcavities, but so far has eluded observation. Here, we experimentally demonstrate phonoritons, when two exciton-polariton condensates confined in a $\mu m$-sized trap within a phonon-photon microcavity are strongly coupled to a confined phonon which is resonant with the energy separation between the condensates. We realize control of phonoritons by piezoelectrically generated phonons and resonant photons. Our findings are corroborated by quantitative models. Thus, we establish zero-dimensional phonoritons as a coherent microwave-to-optical interface.

One strategy towards control of opto-electronic phenomena at the nano- and ps-scale, interconversion of optical and microwave photons for communication between distant qubits[1–3] as well as optical information transfer in on-chip computational devices[4,5] uses optomechanical interactions[6], i.e., correlations between optical and mechanical degrees of freedom. In this setting, optomechanical systems relying on the coupling between high-frequency vibrations (e.g., GHz phonons) and solid-state excitations have become relevant for advanced photonic applications, including the emerging fields of quantum sensing, communication[7] and control of various quantum states[8–14], e.g., qubits[15].

In general, coherent interactions between photons and phonons require a large coupling energy as well as low phonon ($\Gamma_M$) and photon ($\gamma_{phot}$) decoherence rates. This regime, which is a staple for interconversion, has been realized in a variety of optomechanical setups

utilizing direct photon-phonon coupling[7]. High-efficiency transduction between the particles presupposes a single-photon optomechanical cooperativity

$$C_0 = \frac{4 \times g_0^2}{\gamma_{phot} \times \Gamma_M} \tag{1}$$

exceeding unity[6], where $g_0$ is the single-photon optomechanical coupling rate. The coupling rate can be enhanced by the photon population ($N_{phot}$) as $g = g_0 \times \sqrt{N_{phot}}$. Furthermore, in order to realize complete quantum control including interconversion and storage, one requires optomechanical strong coupling (OSC) between the photon and phonon: $g > \{\gamma_{phot}, \Gamma_M\}$[6]. Due to the stringent experimental conditions, only a few systems have reached the OSC regime, which include, e.g., microwave cavities at millikelvin temperatures[16], levitated

[1]Paul Drude Institute for Solid State Electronics, Leibniz Institute in the Research Association Berlin e. V., Hausvogteiplatz 5-7, 10117 Berlin, Germany. [2]Bariloche Atomic Centre and Balseiro Institute, National Council for Scientific and Technical Research, 8400 S.C. de Bariloche, R.N., Argentina. [3]Institute of Nanoscience and Nanotechnology, National Council for Scientific and Technical Research, 8400 Bariloche, Argentina. [4]Department of Applied Physics II, University of Seville, E-41012 Sevilla, Spain. ✉e-mail: kuznetsov@pdi-berlin.de

particles in a cavity[17,18], micromechanical oscillators[19] and Fabry-Perot cavities[20].

The realization of the OSC regime in solid-state systems is highly desirable due to the possibility of harnessing high and ultra-high mechanical frequencies in the GHz and THz ranges, coherent microwave-to-optical interconversion as well as prospects for miniaturization and scalability. However, there are several challenges imposed by the typically low values of $g$ relative to $\Gamma_M$ or $\gamma_{phot}$, the huge mismatch between the phonon ($f_M$) and the photon ($f_{phot}$) frequencies $f_M \ll f_{phot}$, and dissimilar spatial dimensions (wavelengths) of the optical and phonon modes, as well as their non-vanishing dissipative rates. In this context, polaromechanical systems – optomechanical setups utilizing strongly coupled excitons and photons (simply, polaritons[21] or MPs) in semiconductor microcavities (MCs) – become an attractive option[22]. Such MCs can simultaneously confine photons and phonons within their spacer region embedding quantum wells and benefit from the large deformation potential-mediated exciton-phonon coupling[23,24]. Being composite bosons, polaritons can undergo a transition to a non-equilibrium Bose-Einstein condensate (BEC), forming a state described by a macroscopic wavefunction with long spatial and temporal coherences. The BEC temporal coherence ($1/\gamma_{MP}$) can reach the $ns$-range, making them attractive for GHz optomechanics in the coherent (non-adiabatic) regime, i.e., $\gamma_{MP} < f_M$[25,26]. Owing to the matter component, the polariton-phonon coupling rate is significantly larger than that of photons[25–29]. In conjunction with the simultaneous phonon and polariton confinement, this opens the way for the near-unity single-polariton optomechanical cooperativity[30].

Access to such enhanced polariton-phonon interaction regime could enable the optomechanical strong-coupling regime. Around 1982 Keldysh and Ivanov[31,32] have theoretically considered the propagation of exciton-polariton waves in a direct band gap semiconductor crystal. They showed that the interactions between the polariton waves and longitudinal acoustic phonons can lead to light-matter-sound quasiparticles – the phonoritons – arising from the strong coupling between photons, phonons and excitons. Conceptually, due to the approximately four orders of magnitude difference between polariton and phonon energies, phonoritons require two polariton states whose energy difference matches the phonon energy. Therefore, a phonoriton can be considered as a "dressed" polariton that resonantly emits and absorbs phonons as it propagates. The above contrasts phonoritons with conventional exciton-polaritons and phonon-polaritons, which are quasiparticles arising from the *resonant* coupling between photons and excitons, and between infrared photons and transverse optical phonons, respectively. Phonoritons eluded in-depth investigation due to the stringent experimental conditions requiring high-intensity resonant laser beams that complicate optical detection. The existence of phonoritons was suggested in early experiments, which required optical densities in the $10 - 10^2$ MW/cm$^2$ range[33–35]. More recently, Latini et al.[36] predicted the emergence of phonoritons in a MC with an embedded monolayer of h-BN due to the strong-coupling of two exciton-polariton resonances with phonon replicas, which is tunable by the cavity-matter coupling strength. These phonon-exciton-photon states of matter are relevant, e.g., for emerging optomechanical schemes for frequency interconversion[30], phonon and photon lasing[26], acoustic diodes[37] and the enhancement of high-temperature superconductivity[38].

In this work, we experimentally demonstrate MC phonoritons resulting from the strong coupling between two polariton BEC modes with an energy separation equal to confined phonon quanta. For this purpose, we utilize the setup schematically shown in Fig. 1a, where polaritons and GHz phonons are confined in three dimensions within a $\mu$m-sized trap created in the MC spacer region by patterning[26,39,40]. As will be discussed in detail in the next section, two longitudinal (LA) and transverse (TA) polarized phonon modes with frequencies $f_{TA} \approx 2 \times f_{LA}$ are confined in the same trap. Figure 1b summarizes the main results.

Firstly, a non-resonant continuous wave laser focused on the trap excites polaritons in the ground (GS) and excited (ES) states of the trap, which are well-separated in energy: $\Delta_{GS-ES} \approx 20 \times hf_{TA}$. By increasing the laser power ($P_{Exc}$) above the polariton condensation threshold ($P_{Ths}$), GS polaritons transition to the BEC state with linewidth $\gamma_{MP} < \gamma_{LA}$ satisfying the condition for the non-adiabatic (resolved-sideband) modulation regime for LA and TA phonons. We show that BEC GS splits into two pseudo-spin components with an energy splitting $\Delta E_{\uparrow\downarrow}$. With increasing optical excitation, this splitting locks to the TA-phonon energy (i.e., $\Delta E_{\uparrow\downarrow} = hf_{TA}$) and the higher-energy pseudo-spin mode splits by a small energy difference $\delta_{\uparrow\uparrow}$, as indicated in RF Off section of Fig. 1b. We assign the locking and the secondary splitting to the OSC between the pseudo-spin states with self-induced TA-phonons leading to the formation of phonoritons. The phonoriton-related splitting is conceptually similar to the theoretically predicted one in ref. 36. A deformation potential interaction model predicts a phonon-induced interaction energy between the pseudo-spins $g_{\uparrow\downarrow} \geq \{\gamma_{MP}, \Gamma_M\}$, thus fulfilling the condition for the phonoriton formation. In the phonoriton regime, we also observe LA-phonon self-oscillations (SOs), i.e., excitation of coherent population of LA-phonons by the phonoriton field. Secondly, for a fixed $P_{Exc}$, we demonstrate that we control the strength ($g_2$) of the two LA-phonon-coupling between the pseudo-spin states by tuning the population of LA-phonons using a piezo-electric transducer, cf. RF On section of Fig. 1b. By increasing the LA-phonon amplitude (proportional to the RF power, $P_{RF}$, applied to the transducer), we demonstrate tunable LA-phonon sidebands as well as a reduction of the linewidth of the sidebands to precisely 1/2 of their original value, which is an evidence for the RF-induced crossover to the phonoriton regime. Finally, we demonstrate the coherent optical control of $f_{LA}$-phonoritons using a resonant laser beam. The implications of these results for the bidirectional optical-to-microwave interface and coherent control for the quantum regime are discussed.

## Results

### Coherent polaromechanical platform

The studies were carried out using a hybrid (Al,Ga)As MC with acousto-optical distributed Bragg reflectors (DBRs) designed to provide an out-of-plane confinement, i.e., along the MC growth axis ($z\|[001]$), for $\lambda_c = \lambda \times n_c = 810$ nm photons, where $n_c$ is the average refractive index of the MC spacer as well as acoustic phonons with wavelengths of $3\lambda$ and $\lambda$ [cf. Fig. 1(a)]. A detailed description of the MC structure, as well as simulations of its optical and acoustic modes are provided in the supplement notes SM-I-A and SM-I-B, respectively.

The relevant phonon modes have either longitudinal (LA) or transverse (TA) preferential polarizations with frequencies $f_{LA}^{(3\lambda)} = 7$ GHz and $f_{LA}^{(\lambda)} = 20$ GHz for the LA modes and $f_{TA}^{(i\lambda)} \approx 0.7 \times f_{LA}^{(i\lambda)}$ for the TA ones (SM-V-B), where $i = 1, 3$ is the order of the mode. Here, the constant 0.7 is the ratio between the TA and LA sound velocities along z-axis. The above implies that $f_{TA}^{(\lambda)} \approx 2 \times f_{LA}^{(3\lambda)}$. The MC was designed to maximize the overlap between photon and QW excitonic fields as well as between excitons and the strain field of the $3\lambda$-phonons. Nevertheless, the resulting strain amplitudes of the $\lambda$-phonons at the QWs exceed (by approximately one order of magnitude) the ones for the $3\lambda$-mode due to the much higher acoustic reflectivity of the DBRs for $\lambda$-phonons. The MC layer structure also maximizes the coupling between phonons and polaritons via the deformation potential of the GaAs QWs[29]. The cavity photon strongly couples to the heavy-hole and light-hole excitons with coupling strengths of 6 meV and 2 meV respectively. The light-matter properties of the cavity are summarized in SM-II-A.

The MC spacer is photolithographically patterned to produce nm-high and $\mu$m-wide mesas within its spacer to define traps with zero-dimensional confinement for both polaritons and phonons[41]. Finally, a ring-shaped piezoelectric bulk acoustic wave resonator (BAWR) was fabricated on the top surface of the MC to electrically inject highly monochromatic LA phonons with the frequency $f_{BAW} = f_{LA}^{(3\lambda)}$ into the

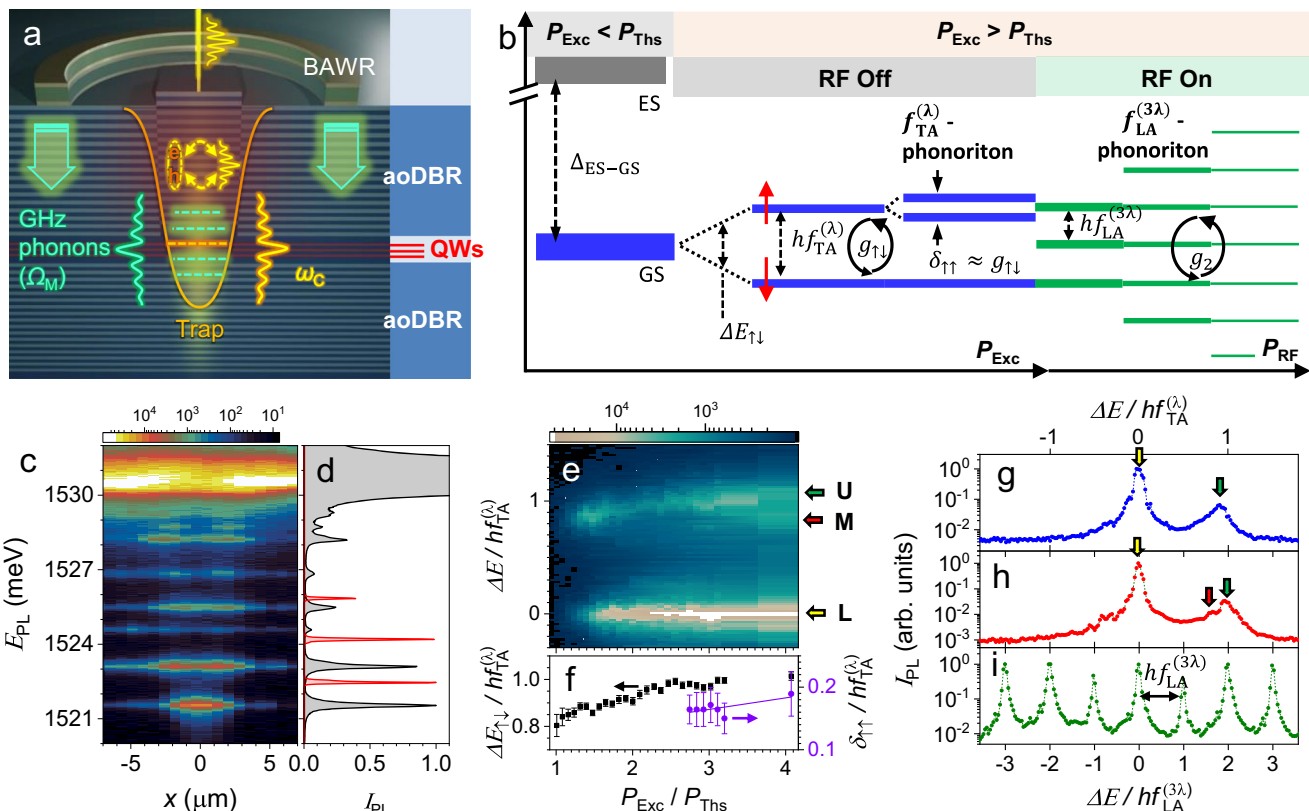

**Fig. 1 | Microcavity phonoritons. a** Sketch of a structured MC, which consists of a spacer embedding quantum wells (QWs) sandwiched between acousto-optic distributed Bragg reflectors (aoDBRs). The $\mu m$-wide and nm-high mesa within the spacer provides lateral confinement potential (the trap depicted by the yellow curve) for polaritons and phonons. The latter are injected optically or using a ring-shaped piezoelectric bulk acoustic wave resonator (BAWR). The phonons non-adiabatically modulate the discrete polariton energy levels (horizontal dashed yellow line) to form sidebands (dashed green lines). **b** Schematic representation of the relevant energy diagrams realized in the experiment. Full description is in the text. $P_{Exc}$ is the optical excitation power; $P_{Ths}$ is the condensation threshold power; GS and ES designate the trap ground and excited state, respectively; RF Off and RF On designate the conditions with the radio-frequency driving of the BAWR off and on, respectively. **c** Spatially and energy-resolved emission spectrum of trap $T_2$ under weak non-resonant excitation. **d** Spatially integrated spectra of the same trap below the threshold (black line) and in the BEC regime (red line). **e** Spectrum of trap $T_1$ GS as the function of the optical excitation power ($P_{Exc}$) normalized to the threshold power ($P_{Ths} = 60$ mW). The levels are designated $L$, $M$ and $U$ for the lower, middle and upper one, respectively. **f** The splitting between the $L$ and $U$ states ($\Delta E_{\uparrow\downarrow}$) and between the $M$ and $U$ states ($\delta_{\uparrow\uparrow}$) as the function of $P_{Exc}/P_{Ths}$. Profiles of the map in (**e**) for $P_{Exc}/P_{Ths} = 1.8$ in (**g**) and $P_{Exc}/P_{Ths} = 2.9$ in (**h**). **i** Spectrum for $P_{Exc}/P_{Ths} = 2.9$ and under piezoelectric excitation of phonons with the frequency $f_{LA}^{(3\lambda)}$. The error bars in (**f**) correspond to the standard error of the Gauss function used to fit the corresponding peaks.

trap. This ringed-shape design of the active (phonon-generating) region of the BAWR allows optical access to the cavity while concentrating the acoustic strain of the injected phonons towards the trap located at the center of the aperture[29], as schematically shown in Fig. 1a and further in Fig. SM-4a,b.

We will present experimental results recorded on two square polariton traps, designated $T_1$ and $T_2$, with the side length $a = 4\,\mu m$ and polariton excitonic contents given by the Hopfield coefficient $X^2 = 0.05$ (corresponding to a detuning $\delta_{CX} = -10$ meV between the bare photon and exciton energies) and $X^2 = 0.2$ ($\delta_{CX} = -5$ meV), respectively.

Figure 1 c shows a typical spatially and energy-resolved photo-luminescence (PL) spectrum of a trap recorded at optical excitation power ($P_{Exc}$) below the condensation threshold (and without external phonon injection). The spatially integrated PL spectrum of the trap shown in Fig. 1d by the black line depicts a discrete energy spectrum typical for a particle in a box. We note that the energy separation between the GS and the first excited state far exceeds $hf_{TA}^{(\lambda)}$. The transition to the BEC regime at high $P_{Exc}$ is accompanied by an energy blueshift and a nonlinear increase of the PL intensity as well as line-width narrowing (cf. red line in Fig. 1d and further details in SM-II-B). The high-resolution spectrum of the trap $T_1$ ground state (GS) in the BEC regime for $P_{Exc} \approx 1.8 \times P_{Ths}$ is shown Fig. 1g. Here and in the following sections, the high-resolution spectra are plotted relative to the

zero-phonon line (ZPL). The emission is dominated by the strong line at zero, with a linewidth (defined as the full width at half maximum) of $\gamma_{MP} \approx 1.3$ GHz $\ll f_{LA}^{(3\lambda)}$, which enables reaching the non-adiabatic interaction regime with the LA and TA phonons. The weaker peak displaced by $f_{TA}^{(\lambda)}$ will be discussed in the next section. For completeness, the dependence of $\gamma_{MP}$ on $P_{Exc}$, detailed in SM-III-B, shows that the line-width reaches sub-GHz values down to $\gamma_{MP} \approx 0.5$ GHz. The linewidth for phonons was determined from power reflection parameter ($s_{11}$) measurements, as described in ref. 29. LA-phonon linewidth is about 300 smaller than that of the polariton BEC reaching $\Gamma_M \approx 3$ MHz at 10 K. The TA phonons are assumed to have a similar linewidth.

**Phonoritons and optomechanical self-oscillations in single traps**
The color map of Fig. 1e shows the dependence of the GS spectrum of trap $T_1$ on the excitation power referenced to the condensation threshold ($P_{Exc}/P_{Ths}$). For clarity, the spectra for different $P_{Exc}/P_{Ths}$ were shifted to have the same zero (the as-measured data is shown in Figs. SM-3c and SM-III-B). The first remarkable feature is the splitting of the GS into its two pseudo-spin components, denoted $L$ and $U$, with the linewidth $\gamma_L \approx 1$ GHz and $\gamma_U \approx 2$ GHz, respectively. The GS degeneracy can be lifted, e.g., by a small lateral asymmetry of the trap[41], which splits the energy of bare MC photons with different polarizations – the so-called longitudinal-transverse pseudo-spin splitting[42], see also

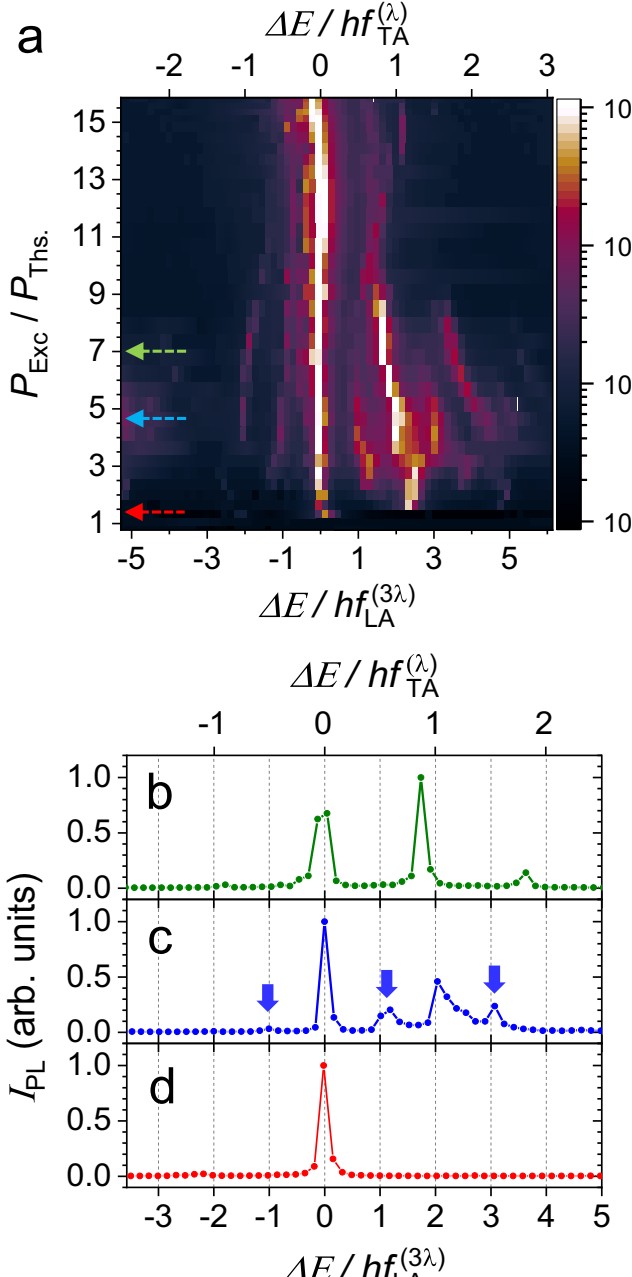

**Fig. 2 | Phonoriton-related self-oscillations. a** Spectral PL map of the GS of $T_2$ for increasing optical excitation power normalized to the threshold power ($P_{Exc}/P_{Ths}$), where $P_{Ths} = 70$ mW. The curves in **b**, **c** and **d** are cross-sections indicated by the arrows in **a** corresponding to $P_{Exc}/P_{Ths} = 1.5$, 4.5 and 7, respectively. The blue arrows in **c** indicate the $f_{LA}^{(3\lambda)}$ sidebands due to the self-oscillations.

SM-V-A and SM-V-B. In the BEC state, the splitting can be amplified by a population mismatch between the pseudo-spins states via polariton-polariton interactions. In fact, Fig. 1e and f show that the magnitude of the energy splitting ($\Delta E_{\uparrow\downarrow}$) increases linearly with $P_{Exc}$ from $\Delta E_{\uparrow\downarrow} = 0.8 \times hf_{TA}^{(\lambda)}$ to $\Delta E_{\uparrow\downarrow} = hf_{TA}^{(\lambda)}$ in the $1 < P_{Exc}/P_{Ths} < 2.4$ range. For higher $P_{Exc}$, however, the splitting locks to the TA-phonon energy, i.e., $\Delta E_{\uparrow\downarrow} = hf_{TA}^{(\lambda)}$. Concurrently, another weaker peak (designated as $M$) appears slightly below the $U$ peak. Figure 1g and h show exemplary profiles of the spectra for $P_{Exc}/P_{Ths} = 1.8$ and $P_{Exc}/P_{Ths} = 2.9$, respectively. The circles in Fig. 1f show that the magnitude of this secondary splitting $\delta_{\uparrow\uparrow}$ increases with $P_{Exc}$ from $\delta_{\uparrow\uparrow} = 0.16 \times f_{TA}^{(\lambda)} \approx 2.2$ GHz to $\delta_{\uparrow\uparrow} = 0.19 \times f_{TA}^{(\lambda)} \approx 2.7$ GHz.

As will be justified at the end of this section, the phonon-induced coupling ($g_{\uparrow\downarrow}$) between the pseudo-spin states can become comparable to the linewidth of the polariton lines for high polariton populations, thus satisfying the condition for the creation of the hybrid phonon-exciton-photon quasiparticle - the phonoriton. Furthermore, the appearance of the $M$-peak in Fig. 1e provides a further experimental signature for the phonoriton formation. In fact, such an energy splitting is presented as the main evidence for phonoriton excitation in the theoretical proposal by Latini et al.[36]. Note that in this proposal the phonon-coupled states are the upper and lower polariton branches, while, in the present work, the relevant polariton states are polariton pseudo-spin states of the trap GS with an energy splitting matching the phonon energy. Phenomenologically, the acoustic modulation creates phonon replicas of the pseudo-spin states. When one of the states anti-crosses the phonon replica of the other state, two lines with a small energy splitting are created. In the present case, an anti-Stokes phonon-replica of the $L$-state couples to the $U$-state when $\Delta E_{\uparrow\downarrow} \approx f_{TA}^{(\lambda)}$, leading to the splitting evidenced by the appearance of an additional line – $M$. The splitting is thus consistent with the theoretical picture described ref. [36]. We note that a similar splitting should also be observed for the $L$ pseudo-spin state. We argue that the observation of this splitting is obscured by the large population ratio (of approx. 50) between the $L$ and $U$ modes, which makes the phonon replicas of the $U$ mode much weaker than the intensity of the $L$ mode. Finally, we note that the experimental values of the coupling $\delta_{\uparrow\uparrow}/2 = 1.25 \pm 0.15$ GHz is comparable to the linewidth of $L$ and $U$ modes. We will present later a model for the interactions between confined polariton states and TA and LA phonons which yields $g_{\uparrow\downarrow}$ on the order of $\delta_{\uparrow\uparrow}/2$.

Interestingly, the emergence of the TA-phonoriton can be accompanied by the $f_{LA}^{(3\lambda)}$-phonon sidebands. Figure 2a shows the spectral PL map of the GS of the more excitonic trap $T_2$ recorded in the BEC regime for increasing $P_{Exc}$. The measurement was carried out with lower resolution in comparison to Fig. 1e and the linewidths are somewhat larger in trap $T_2$, which likely masked the signatures of the secondary splitting. Here, we again observe the pseudo-spin splitting as well as the locking to $hf_{TA}^{(\lambda)}$ in the $3 < P_{Exc}/P_{Ths} < 5$ range. Unlike the case of the more photonic trap $T_1$, in the locking-range, sidebands separated by multiples of $f_{LA}^{(3\lambda)}$ emerge. The sidebands are indicated by the blue arrows in the exemplary profile for $P_{Exc}/P_{Ths} = 4.5$ in Fig. 2c, signalizing phonon self-oscillations (SOs) – the excitation of a coherent mechanical motion by a time-independent polariton drive (also phonon lasing[26]), where the backaction from $f_{LA}^{(3\lambda)}$-phonons leads to the sidebands.

SOs are ubiquitous in optomechanics:[6,43]. In polariton systems, they have been reported for processes of optoelectronic[44] and optomechanical[40] nature. In contrast to the former, the SOs demonstrated here involve transitions between the GS pseudo-spin states rather than between confined levels with different orbitals and larger energy separation. Furthermore, unlike our previous report[26] – in the present case, SOs emerge in a single trap rather than in an array of coupled traps. Importantly, the SOs demonstrate that the phonoriton population can be used to tune the $f_{LA}^{(3\lambda)}$-phonon population. The absence of LA-sidebands in the emission of trap $T_1$ is related to the lower polariton excitonic content in comparison to trap $T_2$, and therefore the reduced coupling to phonons[30].

We can estimate the optomechanical coupling rate ($g^{SO}$) leading to the $f_{LA}^{(3\lambda)}$-sidebands by taking into account that the amplitude of the $n^{th}$ sideband is proportional to $J_n^2(\chi)$, where $J_n$ is the Bessel function of the $n^{th}$ order[45]. The dimensionless modulation index ($\chi$) can be expressed as $\chi = 2g^{SO}/f_{LA}^{(3\lambda)}$[46]. The ratio of the peak intensities of the sideband at $E/hf_{LA}^{(3\lambda)} = 1$ and the ZPL of $J_1^2/J_0^2 \approx 0.3$ implies $\chi \approx 1.2$ and hence $g^{SO} \sim 0.6 \times f_{LA}^{(3\lambda)} > \{\Gamma_M, \gamma_{MP}\}$. Therefore, SOs provide another quantitative evidence of the formation of the phonoriton. Interestingly, phonoritons involving $\lambda$ and $3\lambda$ phonons can appear simultaneously, thus, indicating that more than one phonon mode can enter the OSC regime.

In the following, we show that a simple first-order deformation potential interaction between pseudo-spin states mediated by confined phonons provides an optomechanical coupling with the appropriate symmetry and strength leading to the OSC and SOs. Here, we highlight the main results regarding the interactions between the GS phonons and polaritons in the GS and the first ES of a trap. A detailed analysis of the impact of strain on the polariton states can be found in SM-V-A, SM-V-B and SM-V-C. For that purpose, we first state that the traps considered here are characterized by a small lateral asymmetry[41]: $\Delta a/a = (a_x - a_y)/a$, where $a_x$ and $a_y$ are side lengths and $a = (a_x + a_y)/2$. We note that in an asymmetric trap, the LA phonon with strain along the MC growth-direction can induce transverse deformations, as defined in Fig. SM 10. Indeed, a non-vanishing value of $\Delta a/a$ mixes acoustic modes with different polarizations: the three resulting vibrational eigenstates for $3\lambda$-phonons become a pure TA mode, a second transverse mode (TA') with a small longitudinal admixture, and a longitudinal (LA') mode with a small transverse component, cf. Table 1 (the TA' mode is not shown). For small ratios $\lambda/a$, the confined mode frequencies remain very close to the bulk values. In addition, the small anisotropy $\Delta a/a \sim 0.1$ splits the trap GS into two pseudo-spin components with a small energy difference $\Delta E = 0.01 \times hf_{LA}^{(3\lambda)}$, which, as shown in Fig. 1f, may change with the optical excitation power.

The deformation potential interaction between the pseudo-spins of the confined polariton induced by a single-phonon modulates the energy of each pseudo-spin state (indicated with up and down arrows) by $g_{0,\uparrow\uparrow} - g_{0,\downarrow\downarrow}$ (designated as $g_{0,\uparrow\uparrow/\downarrow\downarrow}$ in the table) and also couples states with opposite spins with a coupling factor $g_{0,\uparrow\downarrow}$. These coupling rates are defined by the uniaxial and shear components of the phonon strain field, respectively. Table 1 summarizes values for the coupling energies and resulting optomechanical cooperativities induced by $3\lambda$-phonons in a $4 \times 4\ \mu m^2$ trap determined assuming a zero-detuning between cavity and exciton energies ($\delta_{CX} = 0$), as well as typical polariton and phonon decay rates of $\gamma_{MP} = 1$ GHz and $\Gamma_M = 1$ MHz, respectively. According to the table, only LA' phonons provide the non-vanishing $g_{0,\uparrow\uparrow/\downarrow\downarrow}$ required for the energy modulation and sideband formation. The pseudo-spin coupling is the largest for the pure TA mode: for the LA' mode, this coupling is small and proportional to the trap asymmetry $\Delta a/a$. Large magnitudes of $g_{0,\uparrow\downarrow}$ are required for the efficient phonon-mediated transfer of polaritons between the pseudo-spin states. The last but one column of the Table 1 shows the polariton threshold population $N_{MP}^{(SO)}$ required for SOs of TA- and LA'-phonons, which can be estimated using Eq. (1) with $\gamma_{phot}$ replaced by $\gamma_{MP}$.

The following picture then emerges for the onset of phonoritons and related SOs. (i) The pseudo-spin energy splitting depends on the trap geometry and can change with polariton density to match the phonon energy. The strong dependence of the polariton transfer between the pseudo-spin states (mediated by phonon absorption and emission) on the splitting energy tends to equilibrate the difference in populations leading to the locking[40,47]. (ii) The pseudo-spin splitting can match the LA- or TA-phonon energy triggering the respective phonoriton mode. One can see that for the TA mode, the condition for the OSC is reached for $N_{MP} \geq 8 \times 10^5$, indeed, $g_{0,\uparrow\downarrow} \times \sqrt{N_{MP}} \approx 1$ GHz $\geq \{\gamma_{MP}, \Gamma_M\}$. Such values of $N_{MP}$ are achieved in the experiment as detailed in (SM-II-C). For completeness, these estimates also apply to $\lambda$-phonons. Specifically, for the LA' $\lambda$-mode, the coupling energies are proportional to $(\lambda/a)^2$ and, hence, are reduced by $(1/3)^2$. However, this reduction is partially compensated by the much longer lifetime (and the higher amplitude) of the $\lambda$-phonons (cf. Fig. SM-1), which is not accounted for in Table 1. (iii) In the BEC regime, the cooperativity for the TA- and LA'-phonons can exceed unity, which leads to the self-sustained generation of the coherent phonons, i.e., the SO sidebands. From the table, one sees that TA SOs can form for $N_{MP} \leq 1000$, while the LA' SOs, observed in the more excitonic trap $T_2$ ($X^2 = 0.2$), require much larger populations (on the order $\sim 10^5 - 10^6$ polaritons). These predictions are in qualitative agreement with the data of Fig. 2a. Table 1 explains the absence of $f_{LA}^{(3\lambda)}$ sidebands in the more photonic trap $T_1$, cf. Fig. 2a. We recall that the values in the table are calculated for a trap with the polariton excitonic fraction $X^2 = 0.5$, while for trap $T_1$ GS $X^2 = 0.05$. Hence, LA' SOs in this trap require approximately two orders of magnitude larger $N_{MP}$ value than the one realized in the experiment.

## Electrically stimulated phonoritons

A unique feature of our platform is the ability to electrically inject $f_{LA}^{(3\lambda)}$-phonons into traps using a bulk acoustic wave resonator (BAWR) driven with a radio-frequency (RF) voltage, fabricated on the MC surface (Fig. 1a). The acoustic resonance of the cavity is modulated by the Fabry Perot resonances of the substrate, resulting in $Q > 5000$ acoustic modes, as described in SM-II-D. In this section, we show that the injected phonons can stimulate phonoritons resulting from strongly coupled pseudo-spin states. For that purpose, we chose one of the resonances, for which we observed the largest energy modulation of the polariton energy.

Figure 1i shows the spectrum of trap $T_1$ under the modulation by $f_{LA}^{(3\lambda)} = 7$ GHz phonons excited by the BAWR for the optical excitation identical to Fig. 1h. In comparison, one now observes many well-defined sidebands separated by $f_{LA}^{(3\lambda)}$ and symmetric with respect to the ZPL. This demonstrates the non-adiabatic control of the polariton BEC by the tunable phonon amplitude and the conversion from the microwave to the optical domain. Firstly, we note that the intensity of the $f_{LA}^{(3\lambda)}$-sidebands is much larger than that of the higher-energy pseudo-spin state. Secondly, the linewidth of the sidebands is smaller than that of the ZPL of Fig. 1h. We discuss this further.

The color map of Fig. 3a and cross-sections in Fig. 3b-e show the dependence of the PL spectrum of trap $T_1$ on the normalized phonon amplitude ($A_{BAW}$), which is proportional to the square-root of the nominal RF power applied to the BAWR. The spectra for $A_{BAW} < 0.02$ are dominated by the strong ZPL with a weaker pseudo-spin state locked at $f_{TA}^{(\lambda)}$ (red arrow in Fig. 3b). For $A_{BAW} \approx 0.02$, two symmetric $f_{LA}^{(3\lambda)}$-sidebands appear on either side of the ZPL. The weak pseudo-spin state can be clearly identified up to $A_{BAW} \approx 0.1$ (cf. Fig. 3c). At higher acoustic amplitudes, additional symmetric sidebands emerge, reaching up to $\pm 5 \times f_{LA}^{(3\lambda)}$-sidebands. For the intermediate $A_{BAW}$ values, such as in Fig. 3d, the intensity of the ZPL line becomes strongly suppressed. This suppression is a form of the optomechanically induced transparency.

The solid blue lines in Fig. 3b–e are fits given by a sum of Lorentzians with linewidths ($\delta E$) weighted by squared Bessel functions $J_n^2(\chi)$, where $\chi$ is the dimensionless modulation amplitude (see SM-IV-B). The fits show that the acoustic modulation redistributes the oscillator strength (initially at the ZPL) among the sidebands while conserving the overall PL intensity. Figure 3f demonstrates that $\chi$ increases with $A_{BAW}$. Figure 3g shows the dependence of the fitted sideband linewidths

**Table 1 | Calculated optomechanical coupling for the ground state (GS) and the first excited state (ES) polariton modes induced by GS $3\lambda$ phonons of TA and LA polarizations in a $4 \times 4\mu m^2$ trap**

| BAW mode | $f_{BAW}$ (GHz) | MP mode | $g_{0,\uparrow\uparrow/\downarrow\downarrow}$ (MHz) | $g_{0,\uparrow\downarrow}$ (MHz) | $N_{MP}^{(SO)} = 1/C_0$ | $N_{MP}^{(OSC)}$ |
|---|---|---|---|---|---|---|
| TA | 4.6 | GS | 0 | 1.1 | 300 | $8 \times 10^5$ |
|  |  | ES | 0 | 0.9 | 450 | $1.2 \times 10^6$ |
| LA' | 6.5 | GS | 6.7 | $0.03\frac{\Delta a}{a}$ | $4.7 \times 10^5 (\frac{\Delta a}{a})^{-2}$ | $10^{11}(\frac{\Delta a}{a})^{-1}$ |
|  |  | ES | 5.3 | $0.02\frac{\Delta a}{a}$ | $7.4 \times 10^5 (\frac{\Delta a}{a})^{-2}$ | $10^{11}(\frac{\Delta a}{a})^{-1}$ |

The frequencies ($f_{BAW}$) were determined for $3\lambda$ GaAs phonons and slightly underestimate the ones of the MC spacer, which also includes (Al,Ga)As layers. $\Delta E$ is the energy splitting between the pseudo-spin modes and $\frac{\Delta a}{a}$ the trap asymmetry. The phonon and polariton decay rates required for the determination of the threshold population for self-oscillation $N_{MP}^{(SO)}$ and the threshold for the optomechanical strong-coupling $N_{MP}^{(OSC)}$ were assumed equal to the ones of the GS $f_{LA}^{(3\lambda)}$ mode (see SM-V-D for details).

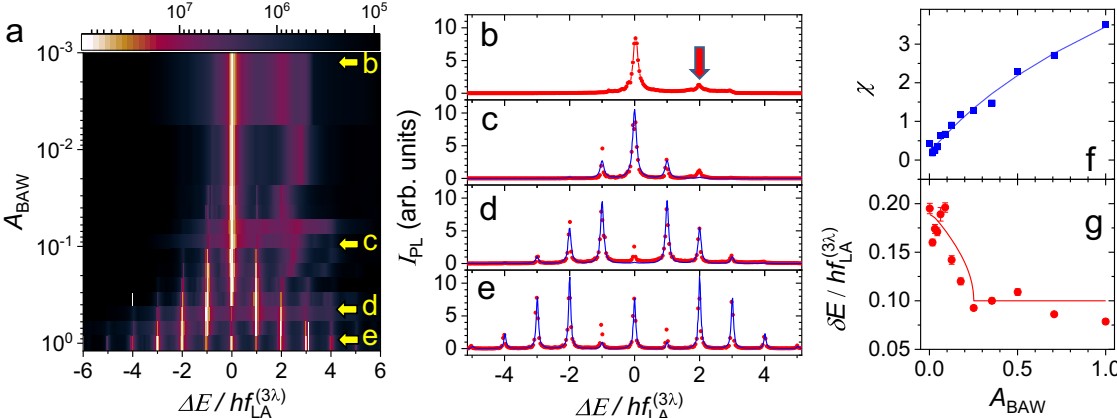

**Fig. 3 | Coherent control of a BEC by the phonon amplitude. a** Dependence of the GS of trap $T_1$ on the acoustic amplitude ($A_{BAW}$) under excitation of the BAWR. **b–e** Cross-sections of **a** along the yellow arrows. Red dots are data and blue solid lines are fits using the model described in the text. The red arrow mark the pseudo-spin-split state. **f** Dependence of the fitted modulation amplitude and **g** linewidth on the normalized acoustic amplitude, $A_{BAW}$. The red solid line represents the prediction of Eq. (3). The blue solid line in **f** is a guide to the eye. The error bars in **g** correspond to the standard error of the Lorentz function used to fit the corresponding peaks.

$\delta E(A_{BAW})$ on $A_{BAW}$. Remarkably, $\delta E(A_{BAW})$ decreases sharply by a factor of two from $\delta E(0.1) = 0.2 \times f_{LA}^{(3\lambda)}$ to $\delta E(0.2) = 0.1 \times f_{LA}^{(3\lambda)}$ and then remains constant. The reduction coincides with the appearance of the first sidebands, i.e., when $\chi \approx 1$, cf. Fig. 3f. A similar linewidth reduction is also observed for the first excited state of the trap (cf. SM-IV-C).

The OSC between particles with largely dissimilar lifetimes leads to quasiparticles with lifetimes approximately twice the one of the shorter-lived component. We assign the observed linewidth reduction to a similar phenomenon. Here, the long-lived phonon field with $\Gamma_M << \gamma_{MP}$ drives coherent oscillations between the pseudo-spin polariton states. In the weak-coupling limit, the states do not swap before decaying with the rate $\gamma_{MP}$, given by the BEC natural linewidth. In the OSC limit, the condensate swaps between the pseudo-spin states at a rate $\gg \gamma_{MP}$. In effect, phonoritons spend half of the time as phonons with $\Gamma_M \ll \gamma_{MP}$, thus leading to a decay rate $\gamma_{MP}/2$.

The first-order coupling between the pseudo-spin states by $f_{LA}^{(3\lambda)}$-phonons is two orders of magnitude weaker than the one by $f_{TA}^{(\lambda)}$-ones and vanishes for perfectly square traps, cf. Table 1. Here, we propose that the coupling by $f_{LA}^{(3\lambda)}$-phonons is enhanced by the large phonon population excited by the BAWR, leading to a quadratic coupling. We use a Hamiltonian (derived in SM-V-E):

$$\hat{H} = \hbar\Omega_M \hat{b}^\dagger \hat{b} + \sum_{i=l,u} \hbar\omega_i \hat{a}_i^\dagger \hat{a}_i + \hat{H}_{int}, \tag{2}$$

where $\hat{a}_i^\dagger$ ($\hat{a}_i$) is the creation (annihilation) operator for a polariton in $i$-mode with energy $\hbar\omega_i$, where i = {l, u} for the lower ($l$) energy and the upper ($u$) energy mode of the pseudo-spin split GS, and $\hat{b}^\dagger$ ($\hat{b}$) is the creation (annihilation) operator of the cavity phonon of energy $\hbar\Omega_M$. The interaction term $\hat{H}_{int} = \hbar G_2 (\hat{a}_u^\dagger \hat{a}_l + \hat{a}_l^\dagger \hat{a}_u)(\hat{b}^\dagger + \hat{b})^2$, in Eq. (2), couples two BEC states separated by $2 \times \hbar\Omega_M$ and is quadratic in phonon amplitude, i.e., $(\hat{b}^\dagger + \hat{b})^2$. A detailed analysis of the interaction, described in SM-V-E, yields a coupling strength $g_2 = 2\sqrt{N_{MP} n_b} G_2$, where $N_{MP}$ is the polariton population and $G_2 = g_{0,\uparrow\uparrow/\downarrow\downarrow} \times g_{0,\uparrow\downarrow}/f_{LA}^{(3\lambda)}$. Importantly, the effective optomechanical coupling depends on the population of phonons, $n_b$, which can be precisely tuned by the BAW amplitude. The compounded mechanical and optical enhancements of the effective coupling constant relaxes the number of phonons and polaritons required for reaching the strong coupling regime. With the interaction, the simplified expression for the eigenfrequencies

becomes $\omega_\pm = -j\frac{(\gamma_l + \gamma_u)}{4} \pm \sqrt{g_2^2 - \frac{(\gamma_l + \gamma_u)^2}{4^2}}$, where $j = \sqrt{-1}$. We assume that $\gamma_l + \gamma_u \approx 2 \times \gamma_{MP}$, then the linewidth, which is the imaginary part of above expression, can be written as:

$$\delta E = \frac{\gamma_{MP}}{2} + Im\left[\sqrt{g_2^2 - \frac{\gamma_{MP}^2}{4}}\right]. \tag{3}$$

Hence, the OSC condition becomes $g_2 \geq \gamma_{MP}/2$, for which the value of $\delta E$ reduces by 50%. This is due to the fact that these excitations are half mechanical and the phonon lifetime, being much larger than $\gamma_{MP}^{-1}$, virtually does not contribute to the linewidth.

The fit to the data using the Eq. (3) normalized to $\gamma_{MP}$ is displayed by the solid line in Figure 3g. In the calculations, we used the experimentally determined polariton and phonon decay rates $\gamma_{MP} = 1.4$ GHz, $\Gamma_M = 3$ MHz, respectively, and an estimated BEC population $N_{MP} = 5 \times 10^6$. The phonon population $n_b$ was determined for each $A_{BAW}$ as described in SM-II-E. The calculations reproduce well the linewidth narrowing to 1/2 of its initial value, confirming the formation of a stimulated phonoriton. However, the fitted $G_2$ value is ~ 40 ± 10 times larger than the one deduced for the pure optomechanical rates in the Table 1. We point out that phonons also affect non-linear polariton interactions[30,48], including those involving the excitonic reservoir[44,49], which are not accounted for in the model. Combined with the optomechanical coupling proposed here, these mechanisms may additionally enhance the polaromechanical coupling.

## Coherent optical control of phonoritons

Finally, we demonstrate optical control of phonoritons in a trap using the setup depicted in Fig. 4a, which is complementary to the mechanical control addressed in the previous section. For that purpose, a weak single-mode control laser with tunable energy $\Delta_L$ was scanned in energy steps of 2.3 GHz across the GS of trap $T_1$. PL spectra were then recorded for each $\Delta_L$ as displayed in Fig. 4b. The weak curved stripes separated by $f_{LA}^{(3\lambda)}$ (indicated by the small yellow arrows) are the phonon sidebands due to the modulation by RF-generated phonons. The weak diagonal feature indicated by the diagonal dashed arrow is the Rayleigh scattering of the control laser as it was scanned from positive to negative values of $\Delta_L$. Interestingly, all emission lines redshift by as much as $\Delta_{rs} = 0.2 \times f_{LA}^{(3\lambda)}$, when the control laser is within their spectral range, i.e., for $|\Delta_L| \leq 5 \times f_{LA}^{(3\lambda)}$. This relatively large red-shift is attributed to a renormalization of the phonoriton GS energy under the increased phonon population induced by the control laser[50].

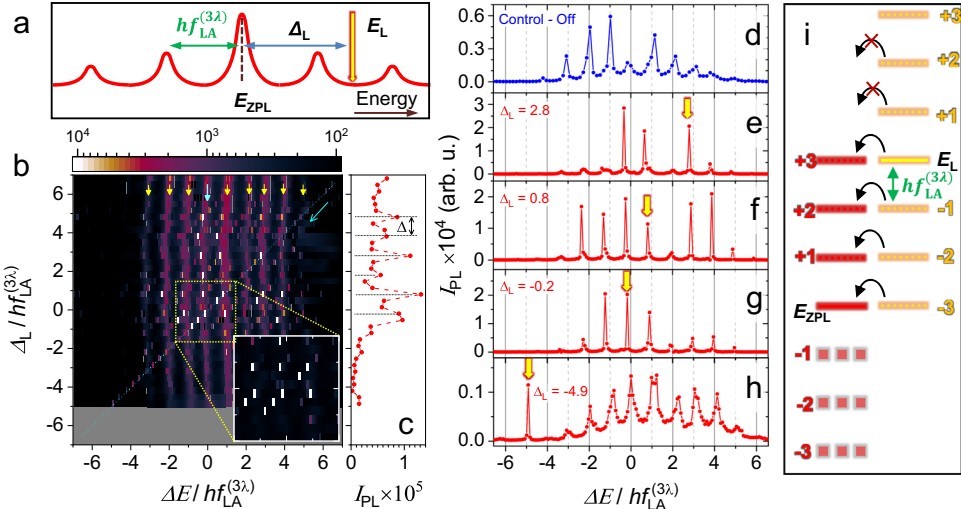

**Fig. 4 | Optical coherent control of phonon sidebands. a** Experiment energy scheme. **b** Spectral dependence of the GS PL of trap $T_1$ on the control laser detuning ($\Delta_L$) under excitation of the BAWR. The tiny white rectangles are the $\delta$-like PL peaks induced by the resonant excitation of the sidebands (yellow arrows). The PL energy (horizontal axis) and $\Delta_L$ (vertical axis) are referenced to the zero-phonon line and scaled to the phonon energy ($hf_{LA}^{(3\lambda)}$). The solid blue arrow indicates the position of the zero-phonon line. The diagonal line indicated by a dashed blue arrow is the Rayleigh scattering from the control laser. The inset shows a magnified region of the map plotted with the intensity in linear scale corresponding to the area defined by the dotted yellow square. **c** Integrated PL intensity of the sidebands as a function of $\Delta_L$. **d** Reference GS spectrum in the absence of the control laser and **e–h** for selected values of $\Delta_L$ indicated by the yellow arrows. **i** An approximate energy diagram for the panel **e**. The two shifted groups of lines represent the phonoriton and the control laser phonon frequency combs.

The most important feature in Fig. 4b is the appearance of $\delta$-like PL peaks whenever the control laser energy matches a sideband, i.e., when $\Delta_L = n_s \times f_{LA}^{(3\lambda)} - \Delta_{rs}$, where $n_s$ is an integer. The latter condition leads to the enhancement of the integrated emission of the sidebands displayed in Fig. 4c. Figure 4e–h show exemplary profiles recorded for $n_s = \{3, 1, 0, -5\}$. At these laser energies, the amplitude of the sidebands increases up to an order of magnitude, while their linewidth reduces to the resolution limit of 0.5 GHz, as compared to the reference spectrum of Fig. 4d recorded without the control laser. This behavior is attributed to the interference of two sets of RF-induced sideband combs: one around the phonoriton ZPL and the second one around the control laser energy. Indeed, as the laser photons enter the MC (and essentially become polaritons), the second frequency comb appears around the laser energy $E_L$ and moves with the latter. An exemplary spectrum for $\Delta_L = +4.5$ showing the two combs is displayed in Fig. SM-9.

Figure 4 i shows a simplified energy diagram describing the interaction of the two combs in the case of Fig. 4e. In the diagram, the group of horizontal lines on the left represents the phonoriton comb, while the one on the right corresponds to the comb of the control laser. The ZPL of the phonoriton and the laser energy are depicted by solid lines, while the dotted lines depict the $f_{LA}^{(3\lambda)}$ sidebands. The thickness of the lines represents their linewidth, while the color saturation represents the emission intensity, with muted colors corresponding to the weaker emission. We also note that the spectrum without the control laser corresponds to the modulation index $\chi \approx 2$ (cf. Fig. 4d). This means that only up to $\pm 3$ sidebands are mainly populated.

Like in the actual case, cf. Fig. 4e, the laser is shown to be detuned by $\Delta_L = +3 \times hf_{LA}^{(3\lambda)}$ with respect to the phonoriton ZPL. The diagram shows that the laser ZPL and its -1, -2 and -3 sidebands are in resonance with the +3, +2 and +1 as well as the phonoriton ZPL, respectively. Under this condition, laser photons are resonantly transferred to the phonoriton sidebands. We point out that the control laser has a nominal linewidth of ~300 kHz, which is ~$10^4$ smaller than the phonoriton one. This explains the observed reduction of the linewidth of the enhanced sidebands. The phonoriton sidebands with indices > +3 are weak (not shown in the diagram), therefore there is a reduced transfer from the laser +1, +2 and +3 sidebands, hence no enhancement

of the emission of the former. Note, however, that not all of the sidebands are enhanced by the same amount, e.g., in Fig. 4e, the +2 sideband remains weak. This indicates that not only the resonant condition between the two combs is important, but also the relative phases between the resonant sidebands, i.e., there can be constructive and destructive interference between the matched sidebands. We point out a general similarity of the realized scheme to the interference of optical sidebands induced by acoustic waves with different frequencies reported for a single QD in ref. 51.

## Discussion

In summary, we have demonstrated that patterned hybrid MCs with traps for polaritons and GHz phonons host light-matter-sound solid-state quasiparticles – phonoritons. The latter arise from the strong coupling between two highly coherent polariton condensates and a phonon mode with energy equal to the detuning between the condensates under lateral confinement in a $\mu m$-large trap. The emergence of phonoritons has been experimentally verified by observing (i) the locking of the two polariton modes to the TA phonon energy accompanied by an additional splitting of one of the polariton modes with the coupling strength tuned by the polariton population (proportional to the optical excitation power); and (ii) the 1/2-narrowing of the condensate linewidth resulting from the enhancement of the quadratic coupling strength by the LA-phonon population. The latter demonstration takes advantage of a unique ability to precisely control the phonon amplitude using piezoelectric acoustic bulk transducers. Furthermore, the extracted magnitudes of couplings match well the ones obtained using quantitative models. The latter provide a background to understand the findings. The polariton-phonon coupling leads to enhanced optomechanical cooperativity that reaches values of $10^4$, resulting in phonon self-oscillations (phonon lasing). We also demonstrated that the phonoriton spectrum can be controlled using an external resonant laser beam. The demonstrated platform for MC phonoritons is promising for the coherent conversion between microwave and optical domains.

The polaromechanical platform opens the way for GHz phonoritonics. In addition to the optical generation of GHz phonons and the coherent microwave-optical interconversion, we envision that the

polaromechanical platform will be attractive for other conventional and emerging applications. Examples include the amplification of optical signals, the generation of tunable (and symmetric) optical frequency combs for atomic clocks, high precision spectroscopy, optical synthesizers as well as for the preparation of quantum states[52].

Our results also hint at a far richer and previously unexplored physics, which paves the way for new interaction regimes between optical, electronic, and mechanical degrees of freedom in the solid state and challenge the existing understanding of the polaromechanical interactions, in particular, regarding the role of non-linear interactions. The large amplitudes of electrically generated GHz strain open the way to phonon nonlinearities, which can be applied for harmonics generation and mixing as well as for parametric processes and phonon squeezing[53]. The observed GS splitting into two pseudo-spin states suggests that the acoustic strain can be used as a source of synthetic magnetic fields for polariton-based topological structures. Artificial band structures of polaritons[54] and phonons[55] have been demonstrated to host many topological effects. Excitingly, phonoriton lattices could be used to study hybrid phonon-exciton-photon topology. Furthermore, phonons can facilitate inter-trap tunneling complementary to the typically realized Josephson one[40]. Lastly, a further challenge is to reach single-polariton cooperativities $C_0 \geq 1$ at GHz frequencies by exploiting the large polariton-phonon coupling[22,25,28]. For 20 GHz phonons[26,29], the thermal phonon occupation is $n_{th} \approx 1$ at 1 K, which enables single phonon manipulation at relatively high temperatures. The present structures can already reach $C_0 > 1/20$ (cf. Supplementary Table 5): the current understanding provides pathways to increase $C_0$ by optimizing the trap geometry and material properties. The platform can thus provide coherent control at the single particle level, which can be applied for the generation of non-classical light at GHz rates[56] as well as serve as quantum interfaces between remote polariton qubits[57].

## Methods
### Microcavity sample
In the (Al,Ga)As material system, the sound and light acoustic impedances as well as the ratios between sound and light velocities are almost identical. As a consequence of this "double magic coincidence"[23], an (Al,Ga)As MC designed to confine near-infrared photons also efficiently confines GHz phonons.

Studied MCs consist of the lower and upper distributed Bragg reflectors (DBRs) and the MC spacer region containing six 15 nm-thick GaAs QWs separated by 7.5 nm-thick $Al_{0.1}Ga_{0.9}As$ barriers. The position of the QWs is optimized in order to maximize the coupling to photonic and phononic modes of the MC. The lower and upper DBRs consist of triple pairs of [(58.1 nm) $Al_{0.1}Ga_{0.9}As$/ (63.1 nm) $Al_{0.5}Ga_{0.5}As$], [(58.1 nm) $Al_{0.1}Ga_{0.9}As$/ (67.6 nm) $Al_{0.9}Ga_{0.1}As$] and [(63.1 nm) $Al_{0.5}Ga_{0.5}As$/ (67.6 nm) $Al_{0.9}Ga_{0.1}As$]. The DBR design provides confinement for optical and acoustic modes with wavelengths $\lambda_o \approx 809/n_{GaAs}$ nm and $\lambda_a = 3\lambda_o$, respectively. Here, $n_{GaAs}$ is the GaAs refractive index. The spacer is 3/2$\lambda_o$ cavity for photons. The acoustic wavelength corresponds to bulk phonons of ~ 7 GHz. Optical and phonon response of the MCs is detailed in SM-I-B. The coupling between polaritons and phonons is dominated by the deformation potential interaction.

### Polariton and phonon confinement
Structured MCs are fabricated by interrupting the growth by the molecular beam epitaxy after the deposition of the cavity spacer embedding the QWs and structuring it by photolithographically defined shallow etching[58]. Zero-dimensional confinement regions (the traps) for photons and phonons are defined by nm-high and $\mu$m-wide regions created within the MC spacer[41] (cf. Fig. 1a). We present experimental results recorded on two square polariton traps (traps $T_1$ and $T_2$) with side $a = 4$ $\mu$m and excitonic contents $X^2 = 0.05$ (corresponding to a detuning $\delta_{CX} = -10$ meV between the bare photon and exciton energies) and $X^2 = 0.2$ ($\delta_{CX} = -5$ meV), respectively.

### Bulk acoustic wave transducers
The phonon generation relies on the transduction of a super-high-frequency (SHF, 3–30 GHz) radio frequency (RF) voltage to sound waves achieved in capacitor-like piezoelectric structures – bulk acoustic wave (BAW) resonators (BAWRs)[59]. A ring-shaped piezoelectric bulk acoustic wave resonator (BAWR)[29] was fabricated on the top surface of the MC to inject monochromatic LA BAWs with frequency $f_M$ tunable around $f_{LA}^{(3\lambda)} = 7$ GHz into the trap.

An important feature of SHF BAWs is the very weak and essentially frequency-independent acoustic attenuation at temperatures below ~ 30 K. This leads to exceptionally long BAW propagation lengths, which reach up to a *cm*. Substrates with polished back-surfaces thus become efficient acoustic cavities with enhanced acoustic amplitudes: here, the BAWs experience specular reflection at the surfaces and make several round trips through the MC spacer before they attenuate. This phonon backfeeding to the MC region boosts the effective quality factor ($Q_{a,eff}$) to values $Q_{a,eff} > 10^4$ in the 5–20 GHz range and, hence, to very large $Q_{a,eff} \times F$ products exceeding $10^{14}$.

### Optical characterization
The spatially- and energy-resolved photoluminescence (PL) measurements were carried out at 10 K temperature in a cryogenic (liquid He) cryostat. The condensates were excited using a single-mode continuous wave (cw) cavity-stabilized laser with the wavelength tuned in the 760–780 nm range. The Gaussian-like excitation spot was chosen to have the diameter of ~ 40$\mu$m on the sample and was centered on the trap. The excitation beam had incidence angle of ~ 15°. For the standard measurements with the spectral-resolution of about 0.1 meV, the magnified PL image of the sample was transferred on the entrance slit of a single grating spectrometer and recorded using a Nitrogen-cooled CCD camera.

### High-resolution spectroscopy of condensates
Fig. SM. 3 sketches the high-resolution optical setup used to detect phonon sidebands in the emission of confined condensates. A part of the collected PL was diverted using a mirror and coupled into a single mode (5 $\mu$m core diameter) fiber. A long-pass filter blocked the scattered light from the pump laser. The fiber guided the PL to a piezo-tunable Fabry–Perot etalon (FP). The FP has a finesse of ~ 240 and free spectral range (FSR) of 68 GHz. The transmission wavelength of the FP was tuned by an external voltage source, controlled from a PC. The PL signal filtered by the FP was guided by another single-mode fiber to the entrance of a single grating spectrometer. The latter resolved the PL from the different FSRs of the FP, which was then detected by a nitrogen-cooled CCD. A custom-made software was used to control the voltage applied to the FP, which allowed to conduct scans with a resolution of 0.28 GHz. In order to avoid temperature induced drifts, the FP was actively stabilized with an external heater. In this configuration, we loose the spatial information. In order to avoid collecting PL from other traps on the sample, we measured on the sample area, which contained an isolated trap.

### Resonant optical excitation
Some experiments were carried out with the simultaneous excitation of the trap with two lasers: the non-resonant pump and a control laser. The wavelength of the single mode (linewidth $\gamma_L \leq 300$ kHz) cw control laser was precisely tuned using a feedback signal provided by a high-resolution wavelength meter. The control laser was focused on the same area with the trap into a spot of ~ 40$\mu$m diameter. As schematically shown in Fig. SM 3, the direct reflection of the control laser was blocked.

### Optomechanical coupling in intracavity traps
For the determination of the optomechanical couplings, we first calculated the eigenmodes for photons as well as coupled TA and LA phonons confined in a trap with infinite potential barriers and

dimensions $a_x = a + \Delta a/2$ and $a_y = a - \Delta a/2$ along the $x||[1\bar{1}0]$ and $y||[110]$ directions, respectively (cf. SM-V-A and SM-V-B). For that purpose, the photon (phonon) wave functions were expressed in a basis of sinusoidal orbitals with $p_n$ ($m_n$) lobes along $\eta = \{x, y\}$. The strain field determined from the phonon wave functions was then used to determine the deformation potential coupling to QW excitons using the Pikus and Bir (PB) Hamiltonian[60]. In the last step, we introduced the Rabi coupling between photonic and excitonic modes to determine the polariton eigenstates as well as the optomechanical coupling energies and cooperativities (cf. SM-V-C and SM-V-D). The quadratic interaction leading to stimulated phonoritons was determined by solving the optomechanical Hamiltonian for two states coupled by the phonons. Expressions for the effective quadratic coupling were then determined in the rotation-wave approximation (cf. SM-V-E).

## Data availability

The measurement and numerical simulation data that support the findings within this study are included within the main text and Supplementary Information. The data shown in the figures is essentially the as-measured data. The only post-processing of the data is the relative shift of the spectra in Figs. 1(e), 2(a) and 3(a) so that all spectra in each figure have the same zero. The corresponding original data and the procedure are described in the supplement. All data can be made available upon a request from the corresponding author.

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

## Acknowledgements
ASK and PVS acknowledge the funding from German DFG (grant 359162958) and QuantERA grant Interpol (EU-BMBF (Germany) grant nr. 13N14783). A.A.R and AF acknowledge financial support from the ANPCyT (Argentina) under grant PICT-2018-03255 and the Alexander von Humboldt Foundation. A.A.R. aknowledges support by PAIDI 2020 Project No. P20-00548 with FEDER funds. The authors thank Dr. Stefan Fölsch for discussions and for a critical review of the manuscript as well as the technical support by R. Baumann, S. Rauwerdink, and A. Tahraoui. Data underlying the reported results are included in the main text and supplementary material.

## Author contributions
A.S.K. performed the experiments and, together with PVS, the analysis. K.B. performed the MBE growth of the samples. A.A.R. and A.F. have developed the strong-coupling model explaining the linewidth narrowing. P.V.S. developed the deformation potential model. All authors participated in the discussions. A.S.K. and P.V.S. have conceived the idea, analyzed data and wrote the manuscript with input from the other authors.

## Funding

## Competing interests
The authors declare no competing interests.
