## [Peer Review File · Nature Communications]

REVIEWER COMMENTS

Reviewer #2 (Remarks to the Author):

Kuznetsov et al. characterize the photoluminescence feature of a semiconductor microcavity designed to host exciton-polariton and simultaneously confine acoustic phonons.

The authors observe extra replicas of the exciton polariton resonances in the PL spectra, which they ascribe to the emergence of phonoriton quasiparticles in the system, sustained by the strong coupling between the exciton polariton and the acoustic phonon waves. Additionally, they demonstrate the tunability of the PL spectra by using an acoustic resonator and a tunable laser to drive the PL resonances.

Overall, I find the work well done and interesting for the community. The methods are well described, and the data are presented clearly with adequate modelling.

However, I have some comments for the authors.

1) First and foremost, I could not understand what the authors define as a phonoriton by going through the introduction paragraph. In line 33-39, the phonoritons are introduced as quasiparticles resulting from strong phonon-photon coupling, or at least this is my understanding of what is written. In this sense, I do not understand what distinguish a phonoriton from a phonon-polariton, the latter being a well-known example of strong light-matter interaction between optical phonon and photons. Later in the text, the concept of strong coupling of exciton polariton to phonon is introduced as a strategy to stabilize phonoritons in solid state systems. In some of the cited references, phonoritons are introduced as resulting from photon-exciton-phonon coupling. Is the latter concept the interpretation of the authors as well? It would be helpful to address the problem more precisely in the introduction.

2) Also, it would be helpful to introduce with more details the platform used by the authors to study such phonoritons. In ref. 16 the authors propose to couple exciton polaritons with phonons with an energy matching the Rabi splitting of the polariton branches. In the case of the present paper, what is the foundation of the coupling between the exciton polariton and the acoustic phonons confined in the trap by the authors? I see that detailed calculations for coupling strength are provided in supporting materials, and as one goes through the text the mechanism is unveiled, but it would be helpful for the reader to find a clearer and synthetic message in the main text, possibly earlier on, clarifying the experiment design. For example, at line 121-126 it is proposed that the ground state splitting induced by the asymmetric trap shall match the phonon frequency. All these ingredients could be stated earlier on for helping the reader.

3) At line 65 'Figure 1c' shall be corrected for figure 1b, I believe.

- 4) At line 174-176, how are the decay rates of, respectively, polaritons and phonons determined?
- 5) In fig. 3b), can the author mark which part of the graph is the inset referred to?
- 6) At line 193, does DELTA_rs refers to the redshift? Please state it , if so.
- 7) I did not understand the interference mechanism underlying the selective enhancement of the sidebands discussed in line 202-206. Can the author add a bit more detail and demonstrate how the mechanism they propose match their observations?

I would recommend the publication in Nature Communications after addressing my comments.

Reviewer #3 (Remarks to the Author):

The authors perform spectroscopy on an optical microcavity containing both quantum well exciton, and a trap for acoustic phonons. The authors claim to observe phonon polaritons resulting from the strong coupling between exciton-polaritons and phonons. The results look nice and impressive and these results certainly deserve publication in a good journal. On the other hand, I found the presentation deficient, and that the main claim is not proven by the manuscript as it is organized now.

I did not catch what is the experimental signature of strong coupling, and even worse, what should be this signature, from an experimental point of view ? In 17, the authors argue that strong coupling can take place when a phonon mode is resonant with the energy splitting between a lower exciton-polariton mode and a upper exciton-polariton mode. The computed spectrum shows energy splitting proportional to some coupling as qualitatively expect from a strongly coupled system. It is also what seems to come out from eqs 45-51, page 24, of the supplementary material.

Here, there is no upper mode as far as I understand. The authors give the feeling that two polarization states of the exciton-polariton trap are involved but it is not really said if this is something important or not. I see that the authors observe many regular side modes. Some kind of Raman peaks, coming from both emission and absorption of phonons. There is some control of the emission both by phonon and photon excitations. Polariton lasing, phonon lasing, many things are observed. However I have not been able to understand what was the signature of strong coupling, and even more what it should be from a theoretical point of view. The authors say that cooperativity is large, so it should be strong coupling, and second that they observed a linewidth reduction by factor 2. From my side, I would like to see some splitting of real part of energy.

I think that a good presentation would be to start by summarize what is theoretically expected, showing the simplest equations, and may be a figure, and then to show how this picture is realized in the experiment. I trust that the authors can do that and that what they want to show is present in their

data. I would therefore recommend the resubmission of deeply improved version where strong coupling is clearly, comprehensively demonstrated.

Minor remarks:

Regarding the ref. on Phonoriton, the ref cited (17) is nice. Inside, the phonoriton history is introduced going back to Keldysh in 1979, and then to some experimental works. I would say that it is probably fair at that stage to keep these old references and the rationale telling about the interest of using new systems as the one of ref. 17 and the one used in the present manuscript. This is especially true since there are no strong size constraints in nature comm.

Also in the perspectives, the authors mentioned polariton based topological structures without citations. May be citing a recent review on that topic like *Optical Material Express*, 11, 1119, (2021), would be in order.

Page 2 fig 1.c is called instead of fig .1.b. Ref 23 and 34 are the arxiv and PRL of the same work.

Response to Reviewers.

Reviewer 1 points

We thank the Reviewer for very careful reading of our manuscript and for the comments aimed at improving its impact. We are pleased to find that in the opinion of the Reviewer, the work is *"well done and interesting for the community. The methods are well described, and the data are presented clearly with adequate modelling."*

Finally, the Reviewer writes that *"I would recommend the publication in Nature Communications after addressing my comments."*

Below, we address the comments.

Point 1:

"First and foremost, I could not understand what the authors define as a phonoriton by going through the introduction paragraph. In line 33-39, the phonoritons are introduced as quasiparticles resulting from strong phonon-photon coupling, or at least this is my understanding of what is written. In this sense, I do not understand what distinguish a phonoriton from a phonon-polariton, the latter being a well-known example of strong light-matter interaction between optical phonon and photons. Later in the text, the concept of strong coupling of exciton polariton to phonon is introduced as a strategy to stabilize phonoritons in solid state systems. In some of the cited references, phonoritons are introduced as resulting from photon-exciton-phonon coupling. Is the latter concept the interpretation of the authors as well? It would be helpful to address the problem more precisely in the introduction."

Response:

We thank the Referee for pointing this out. A new paragraph in the introduction on **page 2 (lines 6078)** as well as the updated version of Fig. 1 now explain the meaning of a phonoriton and highlights the difference to exciton-polaritons and phonon-polaritons. The paragraph also describes the history of the phonoriton following the comment of the other Reviewer. Below we include a specific part that addresses the above concern.

*"Access to such enhanced polariton-phonon interaction regime could enable the optomechanical strong-coupling regime. Around 1982 Keldysh and Ivanov [31,32] have theoretically considered the propagation of exciton-polariton waves in a direct band gap semiconductor crystal. They showed that the interactions between the polariton waves and longitudinal acoustic phonons can lead to novel quasiparticles – the *phonoritons* – arising from the strong coupling between photons, phonons and excitons. Conceptually, due to the approximately four orders of magnitude difference between polariton and phonon energies, phonoritons require two polariton states whose energy difference matches the phonon energy. Therefore, a phonoriton can be considered as a "dressed" polariton that resonantly emits and absorbs phonons as it propagates. The above contrasts phonoritons with conventional exciton-polaritons and phonon-polaritons, which are quasiparticles arising from the *resonant* coupling between photons and excitons, and between infrared photons and transverse optical phonons, respectively. Phonoritons eluded in-depth investigation due to the stringent experimental conditions requiring high-intensity resonant laser beams that complicate optical detection. The existence of phonoritons was suggested in early experiments, which required optical densities in the $10\text{--}10^2$ MW/cm² range [33-35]. In contrast, Latini et al. [36] recently predicted the emergence of phonoritons in a MC with an embedded monolayer of h-BN due to the strong-coupling of two exciton-polariton resonances with phonon replicas, which is tunable by the cavity-matter coupling strength. These new phonon-exciton-photon states of matter are relevant, e.g., for novel optomechanical schemes for frequency interconversion [30], phonon and photon lasing [26], acoustic diodes [37] and the enhancement of high-temperature superconductivity [38]."*

Point 2:

”Also, it would be helpful to introduce with more details the platform used by the authors to study such phonoritons. In ref. 16 the authors propose to couple exciton polaritons with phonons with an energy matching the Rabi splitting of the polariton branches. In the case of the present paper, what is the foundation of the coupling between the exciton polariton and the acoustic phonons confined in the trap by the authors? I see that detailed calculations for coupling strength are provided in supporting materials, and as one go through the text the mechanism is unveiled, but it would be helpful for the reader to find a clearer and synthetic message in the main text, possibly earlier on, clarifying the experiment design. For example, at line 121-126 it is proposed that the ground state splitting induced by the asymmetric trap shall match the phonon frequency. All these ingredients could be stated earlier on for helping the reader.”

Response:

We thank the Reviewer for this suggestion. Indeed, in the original submission, the reader was faced with a detailed discussion in the supplement without a high-level overview of the main ingredients and results in the main text. We have addressed this concern by updating the Figure 1, which now schematically shows the main results in the panel b, which is described in the last paragraphs of the introduction (**lines 79-107**). The updated Figure 1 is shown at the end of the response letter. Below we include the new text, which discusses the origin of the phonoriton in the present case.

”In this work, we experimentally demonstrate MC phonoritons resulting from the strong coupling between two polariton BEC modes with an energy separation equal to confined phonon quanta. For this purpose, we utilize the setup schematically shown in Fig. 1a, where polaritons and GHz phonons are confined in three dimensions within a μm -sized trap created in the MC spacer region by patterning [26,39,40]. As will be discussed in detail in the next section, two longitudinal (LA) and transverse (TA) polarized phonon modes with frequencies $f_{\text{TA}} \approx 2 \times f_{\text{LA}}$ are confined in traps. Figure 1b summarizes the main results.

*Firstly, a non-resonant continuous wave laser focused on the trap excites polaritons in the ground (GS) and excited (ES) states of the trap, which are well-separated in energy: $\Delta_{\text{GS-ES}} \approx 20 \times \hbar f_{\text{TA}}$. By increasing the laser power (P_{Exc}) above the polariton condensation threshold (P_{Ths}), GS polaritons transition to the BEC, with linewidth $\Upsilon_{\text{MP}} < f_{\text{LA}}$ satisfying the condition for the non-adiabatic (resolved-sideband) modulation regime for LA and TA phonons. We show that BEC GS splits into two pseudo-spin components with an energy splitting $\Delta_{\text{E}\uparrow\downarrow}$. With increasing optical excitation, this splitting locks to the TA-phonon energy (i.e., $\Delta_{\text{E}\uparrow\downarrow} = \hbar f_{\text{TA}}$) and the higher-energy pseudo-spin mode splits by a small energy difference $\delta_{\uparrow\uparrow}$, as indicated in *RF Off* section of Fig. 1b. We assign the locking and the secondary splitting to the OSC between the pseudo-spin states with self-induced TA-phonons leading to the formation of phonoritons. The phonoriton-related splitting is conceptually similar to the theoretically predicted one in Ref. [36]. A deformation potential interaction model predicts a phonon-induced interaction energy between the pseudo-spins $g_{\uparrow\downarrow} \geq \{ \Upsilon_{\text{MP}}, \Gamma_{\text{M}} \}$, thus fulfilling the condition for the phonoriton formation. In the phonoriton regime, we also observe LA-phonon self-oscillations (SOs), i.e., excitation of coherent population of LA-phonons by the phonoriton field. Secondly, for a fixed P_{Exc} , we demonstrate that we control the strength (g_2) of the two LA-phonon-coupling between the pseudo-spin states by tuning the population of LA-phonons using a piezoelectric transducer, cf. *RF On* section of Fig. 1b. By increasing the LA-phonon amplitude (proportional to the RF power, P_{RF} , applied to the transducer), we demonstrate tunable LA-phonon sidebands as well as a *reduction of the linewidth of the sidebands to precisely 1/2 of their original value*, which is an evidence for the RF-induced crossover to the phonoriton regime. Finally, we demonstrate the coherent optical control of f_{LA} -phonoritons using a resonant laser beam. The implications of these results for the bidirectional optical-to-microwave interface and coherent control for the quantum regime are discussed.”*

In addition, we expanded the description of our platform provided in the section “Coherent polaromechanical platform” on **page 3**.

Point 3:

”At line 65 ‘Figure 1c’ shall be corrected for figure 1b, I believe.”

Response:

We have carefully proofread the text and corrected typos.

Point 4:

"At line 174-176, how are the decay rates of, respectively, polaritons and phonons determined?"

Response:

We added a line on the **page 4 (line 146)** giving the linewidth definition:

"The emission is dominated by the strong line at zero, with a linewidth (defined as the full width at half maximum)."

The determination of the phonon linewidth is described on page **4 (lines 150-153)** with the following text:

"The linewidth for phonons was determined from power reflection parameter (s_{11}) measurements, as described in Ref. [29]. LA-phonon linewidth is about 300 smaller than that of the polariton BEC reaching $\Gamma_M \approx 3$ MHz at 10 K. The TA phonons are expected to have a similar linewidth."

Point 5:

"In fig. 3b), can the author mark which part of the graph is the inset referred to?"

Response:

The following text was added to the Figure 4 (formerly Fig. 3) caption:

"The inset shows a magnified region of the map plotted with the intensity in linear scale corresponding to the area defined by the dotted yellow square".

Point 6:

"At line 193, does DELTA rs refers to the redshift? Please state it , if so."

Response:

The Reviewer is correct, we fixed the sentence on **page 10 (lines 340)**:

"Interestingly, all emission lines redshift by as much as $\Delta r_s = 0.2 \times \frac{f_{LA}^{(3\lambda)}}{f_{LA}}$ ".

Point 7:

"I did not understand the interference mechanism underlying the selective enhancement of the sidebands discussed in line 202-206. Can the author add a bit more detail and demonstrate how the mechanism they propose match their observations?"

Response:

We added a detailed explanation (provided below) of the mechanism for the resonant optical control of phonoritons on the **pages 10-11 (lines 348-375)** as well as an additional panel (i) to the Figure 4 showing the proposed energy diagram. The Figure 4 is included at the end of this response letter.

"This behavior is attributed to the interference of two sets of RF-induced sideband combs: one around the phonoriton ZPL and the second one around the control laser energy. Indeed, as the laser photons enter the MC (and essentially become polaritons), the second frequency comb appears around the laser energy E_L and moves with the latter. An exemplary spectrum for $\Delta_L = +4.5$ showing the two combs is displayed in Fig. SM-9.

Figure 4i shows a simplified energy diagram describing the interaction of the two combs in the case of Fig. 4e. In the diagram, the group of horizontal lines on the left represents the phonoriton comb, while the one on the right corresponds to the comb of the control laser. The ZPL of the phonoriton and the laser energy are depicted by solid lines, while the dotted lines depict the $f_{LA}^{(3\lambda)}$ LA sidebands. The thickness of the lines represents their linewidth, while the color saturation represents the emission intensity, with muted colors corresponding to the weaker emission. We

also note that the spectrum without the control laser corresponds to the modulation index $\chi \approx 2$ (cf. Fig. 4d). This means that only up to ± 3 sidebands are mainly populated.

Like in the actual case, cf. Fig. 4e, the laser is shown to be detuned by $\Delta L = +3 \times hf(\frac{3\lambda}{2\lambda})$ with respect to the phonoriton ZPL. The diagram shows that the laser ZPL and its -1, -2 and -3 sidebands are in resonance with the +3, +2 and +1 as well as the phonoriton ZPL, respectively. Under this condition, laser photons are resonantly transferred to the phonoriton sidebands. We point out that the control laser has a nominal linewidth of ~ 300 kHz, which is $\sim 10^4$ smaller than the phonoriton one. This explains the observed reduction of the linewidth of the enhanced sidebands. The phonoriton sidebands with indices $>+3$ are weak (not shown in the diagram), therefore there is a reduced transfer from the laser +1, +2 and +3 sidebands, hence no enhancement of the emission of the former. Note, however, that not all of the sidebands are enhanced by the same amount, e.g., in Fig. 4e, the +2 sideband remains weak. This indicates that not only the resonant condition between the two combs is important, but also the relative phases between the resonant sidebands, i.e. there can be constructive and destructive interference between the matched sidebands. We point out a general similarity of the realized scheme to the interference of optical sidebands induced by acoustic waves with different frequencies reported for a single QD in Ref. [53].”

Reviewer 2 points

“The results look nice and impressive and these results certainly deserve publication in a good journal.” – We thank the Reviewer for the positive evaluation.

The Reviewer criticises the convoluted presentation of the results and points out that the organization of the original manuscript made it difficult to see the proof of phonoritons: *“On the other hand, I found the presentation deficient, and that the main claim is not proven by the manuscript as it is organized now. I did not catch what is the experimental signature of strong coupling, and even worse, what should be this signature, from an experimental point of view ? ”*

We are thankful to the Reviewer for highlighting this deficiency of the manuscript (and its presentation) as well as for his specific suggestion: *“I think that a good presentation would be to start by summarize what is theoretically expected, showing the simplest equations, and may be a figure, and then to show how this picture is realized in the experiment”.*

We appreciate that the Reviewer sees that our results contain the necessary proof, which is expressed in the following text: *“I trust that the authors can do that and that what they want to show is present in their data.”*

The Reviewer recommends to resubmit an improved manuscript: *“I would therefore recommend the resubmission of deeply improved version where strong coupling is clearly, comprehensively demonstrated.”*

Below, we address the specific comments.

Point 1:

“I think that a good presentation would be to start by summarize what is theoretically expected, showing the simplest equations, and may be a figure, and then to show how this picture is realized in the experiment.”

We find that the above comment is closely related to the following one:

“In 17, the authors argue that strong coupling can take place when a phonon mode is resonant with the energy splitting between a lower exciton-polariton mode and a upper exciton-polariton mode. The computed spectrum shows energy splitting proportional to some coupling as qualitatively expect from a strongly coupled system. It is also what seems to come out from eqs 45-51, page 24, of the supplementary material. Here, there is no upper mode as far as I understand. The authors give the feeling that two polarization states of the exciton-polariton trap are involved but it is not really said if this is something important or not.”

Below we provide an answer to the above concerns.

Response:

Firstly, we have defined phonoritons in the revised abstract:

“Optomechanical systems provide a pathway for the bidirectional optical-to-microwave inter-conversion in (quantum) networks. These systems can be implemented using hybrid platforms, which efficiently couple optical photons and microwaves via intermediate agents, e.g. phonons.

Semiconductor exciton-polariton microcavities operating in the strong light-matter coupling regime offer enhanced coupling of near-infrared photons to GHz phonons via excitons. Furthermore, a new coherent phonon-exciton-photon quasiparticle termed phonoriton, has been theoretically predicted to emerge in microcavities, but so far has eluded observation. Here, we experimentally demonstrate phonoritons, when two exciton-polariton condensates confined in a um-sized trap within a phonon-photon microcavity are strongly coupled to a confined phonon which is resonant with the energy separation between the condensates. We realize control of phonoritons by piezoelectrically generated phonons and resonant photons. Our findings are corroborated by quantitative models. Thus, we establish zero-dimensional phonoritons as a coherent microwave-to-optical interface.”

Secondly, we have enhanced the presentation of the concept of the optomechanical strong coupling (OSC) on **pages 1-2 (lines 36-45)** with the following text:

”The coupling rate can be enhanced by the photon population (N_{phot}) as $g = g_0 \times \sqrt{N_{\text{phot}}}$. Furthermore in order to realize complete quantum control including interconversion and storage, one requires *optomechanical strong coupling* (OSC) between the photon and phonon: $g > \{\gamma_{\text{phot}}, \Gamma_M\}$ [6]. Due to the stringent experimental conditions, only a few systems have reached the OSC regime, which include microwave cavities at millikelvin temperatures [16], levitated particles in a cavity [17,18], micromechanical oscillators [19] and Fabry-Perot cavities [20].

The realization of the OSC regime in solid-state systems is highly desirable due to the possibility of harnessing high and ultra-high mechanical frequencies in the GHz and THz ranges, coherent microwave-to-optical interconversion as well as prospects for miniaturization and scalability.”

And on **page 2 (lines 64-68)** we emphasized the origin and theoretical signatures of the phonoriton:

”Conceptually, due to the approximately four orders of magnitude difference between polariton and phonon energies, phonoritons require two polariton states whose energy difference matches the phonon energy. Therefore, a phonoriton can be considered as a ”dressed” polariton that resonantly emits and absorbs phonons as it propagates.”

and further **(lines 73-76)**

”In contrast, Latini et al. [36] recently predicted the emergence of phonoritons in a MC with an embedded monolayer of h-BN due to the strong-coupling of two exciton-polariton resonances with phonon replicas, which is tunable by the cavity-matter coupling strength.”

The Reviewer correctly points out the similarity between our work and the one from Latini (Ref [36]). Indeed in both works, the phonon energy, which in both cases is approximately four orders of magnitude smaller than that of the exciton and photon, is resonant with the energy splitting between polariton modes. In Latini’s case, the bare polariton modes involved in the phonoriton are the lower and upper polariton branches, which are assumed to have an energy splitting matching the phonon frequency. In our case, the separation between the ground state and the first excited state of the trap (as well as between the lower and upper polariton states) is much larger than the phonon energy. The polariton modes involved in the phonoritons are rather the two pseudo-spin states of the ground state of the trap, which are coupled by the phonon field and have an energy splitting matching the phonon energy, as schematically shown in the updated Fig. 1b presented at the end of the response letter. In order to address this specific question about the polariton modes involved in phonoritons in the present work and in Latini’s proposal, we added the following text on **page 5 (lines 175-180)**:

”Note that in this [AK: Latini’s] proposal the phonon-coupled states are the upper and lower polariton branches, while, in the present work, the relevant polariton states are polariton pseudo-spin states of the trap GS with an energy splitting matching the phonon energy. Phenomenologically, the acoustic modulation creates phonon replicas of the pseudo-spin states. When one of the states anti-crosses the phonon replica of the other state, two lines with a small energy splitting are created.”

We believe that the updated Fig. 1(b) provides a comprehensive high-level overview of the rich palette of the observed phenomena and the relevant energy scales. More importantly, it shows how the coupling

between phonon modes and the pseudo-spin-split levels of the trap ground state lead to the emergence of phonoritons. We added the paragraphs below to the introduction on **page 3 (lines 79-107)**, which summarize the main results and details the updated Fig. 1(b):

”In this work, we experimentally demonstrate MC phonoritons resulting from the strong coupling between two polariton BEC modes with an energy separation equal to confined phonon quanta. For this purpose, we utilize the setup schematically shown in Fig. 1a, where polaritons and GHz phonons are confined in three dimensions within a μm -sized trap created in the MC spacer region by patterning [26,39,40]. As will be discussed in detail in the next section, two longitudinal (LA) and transverse (TA) polarized phonon modes with frequencies $f_{\text{TA}} \approx 2 \times f_{\text{LA}}$ are confined in traps. Figure 1b summarizes the main results.

Firstly, a non-resonant continuous wave laser focused on the trap excites polaritons in the ground (GS) and excited (ES) states of the trap, which are well-separated in energy: $\Delta_{\text{GS-ES}} \approx 20 \times \hbar f_{\text{TA}}$. By increasing the laser power (P_{Exc}) above the polariton condensation threshold (P_{Ths}), GS polaritons transition to the BEC, with linewidth $\gamma_{\text{MP}} < f_{\text{LA}}$ satisfying the condition for the non-adiabatic (resolved-sideband) modulation regime for LA and TA phonons. We show that BEC GS splits into two pseudo-spin components with an energy splitting $\Delta_{\text{E}\uparrow\downarrow}$. With increasing optical excitation, this splitting locks to the TA-phonon energy (i.e., $\Delta_{\text{E}\uparrow\downarrow} = \hbar f_{\text{TA}}$) and the higher-energy pseudo-spin mode splits by a small energy difference $\delta_{\uparrow\uparrow}$, as indicated in *RF Off* section of Fig. 1b. We assign the locking and the secondary splitting to the OSC between the pseudo-spin states with self-induced TA-phonons leading to the formation of phonoritons. The phonoriton-related splitting is conceptually similar to the theoretically predicted one in Ref. [36]. A deformation potential interaction model predicts a phonon-induced interaction energy between the pseudo-spins $g_{\uparrow\downarrow} \geq \{\gamma_{\text{MP}}, \Gamma_{\text{M}}\}$, thus fulfilling the condition for the phonoriton formation. In the phonoriton regime, we also observe LA-phonon self-oscillations (SOs), i.e., excitation of coherent population of LA-phonons by the phonoriton field. Secondly, for a fixed P_{Exc} , we demonstrate that we control the strength (g_2) of the two LA-phonon-coupling between the pseudo-spin states by tuning the population of LA-phonons using a piezoelectric transducer, cf. *RF On* section of Fig. 1b. By increasing the LA-phonon amplitude (proportional to the RF power, P_{RF} , applied to the transducer), we demonstrate tunable LA-phonon sidebands as well as a *reduction of the linewidth of the sidebands to precisely 1/2 of their original value*, which is an evidence for the RF-induced crossover to the phonoriton regime. Finally, we demonstrate the coherent optical control of f_{LA} -phonoritons using a resonant laser beam. The implications of these results for the bidirectional optical-to-microwave interface and coherent control for the quantum regime are discussed.”

The experimental signatures of the strong coupling are addressed in more detail in our response to the Point 2 of the Reviewer (see below).

Point 2:

”I see that the authors observe many regular side modes. Some kind of Raman peaks, coming from both emission and absorption of phonons. There is some control of the emission both by phonon and photon excitations. Polariton lasing, phonon lasing, many things are observed. However I have not been able to understand what was the signature of strong coupling, and even more what it should be from a theoretical point of view. The authors say that cooperativity is large, so it should be strong coupling, and second that they observed a linewidth reduction by factor 2. From my side, I would like to see some splitting of real part of energy.”

Response:

The Reviewer correctly pointed out that the description of the multiple observed optomechanical phenomena was convoluted and did not allow to understand the main phonoriton signatures. As was indicated in the response to the previous comment, we improved the presentation by including a high-level overview (in Figure 1(b) and the last paragraphs of the introduction), which precede the presentation of the actual results.

In terms of the experiment, there are three phenomena relevant for the identification of the phonoriton:

- (i) The existence of two polariton modes (pseudo-spins of the ground state), with the energy separation locked to the TA-phonon energy.

- (ii) (new) The emergence of an additional splitting of the upper-energy pseudo-spin BEC state, when the splitting energy is equal to the TA-phonon energy. The observations (i) and (ii) are similar to the ones theoretically discussed by Latini et al. [36]. The observation of this secondary splitting proves that the system is in the strong optomechanical coupling regime resulting in the formation of the phonoriton quasi-particles.
- (iii) (enhanced) The 1/2 reduction of the spectral linewidth of the LA-phonon sidebands, when the coupling strength between the pseudo-spin split states is increased by increasing the amplitude of phonon mode using the bulk acoustic transducer. This reduction is a signature of the transition to the phonoriton regime, where the strong-coupling between particles with largely dissimilar lifetimes leads to quasiparticles with lifetimes approximately twice the one of the shorter-lived component.

To demonstrate points (i) and (ii), we carefully examined the dependence of the pseudo-spin splitting of the ground state (GS) BEC on the optical excitation power (P_{Exc}), which was previously shown in the supplement figure Fig. SM 6. The dependence is now shown in Fig. 1(e) and 1(f). The text below (included on **pages 4-5 (lines 154-188)** of the revised text) provides the analysis.

”The color map of Fig. 1e shows the dependence of the GS spectrum of trap T_1 on the excitation power referenced to the condensation threshold ($P_{\text{Exc}}/P_{\text{Ths}}$). For clarity, the spectra for different $P_{\text{Exc}}/P_{\text{Ths}}$ were shifted to have the same zero (the as-measured data is shown in the Fig. SM-3c and SM-III-B). The first remarkable feature is the splitting of the GS into its two pseudo-spin components, denoted L and U, with the linewidth $\gamma_L \approx 1$ GHz and $\gamma_U \approx 2$ GHz, respectively. The GS degeneracy can be lifted, e.g., by a small lateral asymmetry of the trap [41], which splits the energy of bare MC photons with different polarizations – the so-called longitudinal-transverse pseudo-spin splitting [42], see also SM-V-A and SM-V-B. In the BEC state, the splitting can be amplified by a population mismatch between the pseudo-spins states via polariton-polariton interactions. In fact, Figs. 1e and 1f show that the magnitude of the energy splitting ($\Delta E_{N\downarrow}$) increases linearly with P_{Exc} from $\Delta E_{N\downarrow} = 0.8 \times \hbar f_{\text{TA}}^{(\lambda)}$ to $\Delta E_{N\downarrow} = \hbar f_{\text{TA}}^{(\lambda)}$

in the $1 < P_{\text{Exc}}/P_{\text{Ths}} < 2.4$ range. For higher P_{Exc} , however, the splitting locks to the TA-phonon energy, i.e., $\Delta E_{N\downarrow} = \hbar f_{\text{TA}}^{(\lambda)}$.

Concurrently, a new weaker peak (designated as M) appears slightly below the U peak. Figures 1g and 1h show exemplary profiles of the spectra for $P_{\text{Exc}}/P_{\text{Ths}} = 1.8$ and $P_{\text{Exc}}/P_{\text{Ths}} = 2.9$, respectively. The circles in Fig. 1f show that the magnitude of this secondary splitting δ_{NN} increases with P_{Exc} from $\delta_{\text{NN}} = 0.16 \times \hbar f_{\text{TA}}^{(\lambda)}$ 2.2 GHz to $\delta_{\text{NN}} = 0.19 \times \hbar f_{\text{TA}}^{(\lambda)}$ 2.7 GHz.

As will be justified at the end of this section, the phonon-induced coupling ($g_{N\downarrow}$) between the pseudo-spin states can become comparable to the linewidth of the polariton lines for high polariton populations, thus satisfying the condition for the creation of the hybrid phonon-exciton-photon quasiparticle – the phonoriton. Furthermore, the appearance of the M-peak in Fig. 1e provides a further experimental signature for the phonoriton formation. In fact, such an energy splitting is presented as the main evidence for phonoriton excitation in the theoretical proposal by Latini et al. [36]. Note that in this proposal the phonon-coupled states are the upper and lower polariton branches, while, in the present work, the relevant polariton states are polariton pseudo-spin states of the trap GS with an energy splitting matching the phonon energy. Phenomenologically, the acoustic modulation creates phonon replicas of the pseudo-spin states. When one of the states anti-crosses the phonon replica of the other state, two lines with a small energy splitting are created. In the present case, an anti-Stokes phonon-replica of the L-state couples to the U-state when $\Delta E_{N\downarrow} \approx \hbar f_{\text{TA}}^{(\lambda)}$

leading to the splitting evidenced by the appearance of an additional line – M. The splitting is thus consistent with the theoretical picture described Ref. [36]. We note that a similar splitting should also be observed for the L pseudo-spin state. We argue that the observation of this splitting is obscured by the large population ratio (of approx. 50) between the L and U modes, which makes the phonon replicas of the U mode much weaker than the intensity of the L mode. Finally, we note that the experimental values of the coupling $\delta_{\text{NN}}/2 = 1.25 \pm 0.15$ GHz is comparable to the linewidth of L and U modes. We will present later a model for the interactions between confined polariton states and TA and LA phonons which yields $g_{N\downarrow}$ on the order of $\delta_{\text{NN}}/2$.”

Moreover, the phonon self-oscillations (SOs) are the consequence of the coupling between the pseudo-spin BEC states and allow an independent estimation of the coupling energy. The following text now found on **page 6 (lines 207-212)** discusses this:

”We can estimate the optomechanical coupling rate (g^{SO}) leading to the $f(3\lambda)$ LA-sidebands by taking into account the fact that the amplitude of the n^{th} sideband is proportional to $J_n^2(\chi)$, where J_n is the Bessel function of the n^{th} order [47]. The dimensionless modulation index (χ) can be expressed as $\chi = 2g^{SO}/f(3\lambda)$ [48]. The ratio of the peak intensities of the sideband at $E/hf(3\lambda) = 1$ and the ZPL of $J_1^2/J_0^2 \approx 0.3$ implies $\chi \approx 1.2$ and hence $g^{SO} \sim 0.6 \times f(3\lambda)$. Therefore, SOs provide another quantitative evidence of the formation of the phonoriton.”

The above experimental findings are corroborated by a theory, which was previously only briefly mentioned in the main text. In the revised version, on **pages 6-7 (lines 215-263)**, we expanded the discussion of the model explaining the origin and magnitudes of the coupling between polariton states and LA- and TA-phonons. The model predicts the splitting of the GS into pseudo-spin components and provides an estimate of the TA-phonon-induced coupling rate between them, see the added text below and the Table I at the end of the response.

”In the following, we show that a simple first-order deformation potential interaction between pseudo-spin states mediated by confined phonons provides an optomechanical coupling with the appropriate symmetry and strength leading to the OSC and SOs. Here, we highlight the main results regarding the interactions between the GS phonons and polaritons in the GS and the first ES of a trap. A detailed analysis of the impact of strain on the polariton states can be found in *SM-V-A, SM-V-B and SM-V-C*. For that purpose, we first state that the traps considered here are characterized by a small lateral asymmetry [41]: $\Delta\mathbf{a}/\mathbf{a} = (\mathbf{a}_x - \mathbf{a}_y)/\mathbf{a}$, where \mathbf{a}_x and \mathbf{a}_y are side lengths and $\mathbf{a} = (\mathbf{a}_x + \mathbf{a}_y)/2$. We note that in an asymmetric trap, the LA phonon with strain along the MC growth-direction can induce transverse deformations, as defined in Fig. SM 10. Indeed, a non-vanishing value of $\Delta\mathbf{a}/\mathbf{a}$ mixes acoustic modes with different polarizations: the three resulting vibrational eigenstates for 3λ -phonons become a pure TA mode, a second transverse mode (TA’) with a small longitudinal admixture, and a longitudinal (LA’) mode with a small transverse component, cf. Table I (the TA’ mode is not shown). For small ratios λ/\mathbf{a} , the confined mode frequencies remain very close to the bulk values. In addition, the small anisotropy $\Delta\mathbf{a}/\mathbf{a} \sim 0.1$ splits the trap GS into two pseudo-spin components with a small energy difference $\Delta E = 0.01 \times hf(3\lambda)$, which, as shown in Fig 1f, may change with the optical excitation power.

The deformation potential interaction between the pseudo-spins of the confined polariton induced by a single-phonon modulates the energy of each pseudo-spin state (indicated with up and down arrows) by $g_{0,\uparrow\downarrow} \sim g_{0,\downarrow\uparrow}$ (designated as $g_{0,\uparrow\uparrow/\downarrow\downarrow}$ in the table) and also couples states with opposite spins with a coupling factor $g_{0,\uparrow\downarrow}$. These coupling rates are defined by the uniaxial and shear components of the phonon strain field, respectively. Table I summarizes values for the coupling energies and resulting optomechanical cooperativities induced by 3λ -phonons in a $4 \times 4 \mu\text{m}^2$ trap determined assuming a zero-detuning between cavity and exciton energies ($\delta_{CX} = 0$), as well as typical polariton and phonon decay rates of $\gamma_{MP} = 1$ GHz and $\Gamma_M = 1$ MHz, respectively. According to the table, only LA’ phonons provide the non-vanishing $g_{0,\uparrow\uparrow/\downarrow\downarrow}$ required for the energy modulation and sideband formation. The pseudo-spin coupling is the largest for the pure TA mode: for the LA’ mode, this coupling is small and proportional to the trap asymmetry $\Delta\mathbf{a}/\mathbf{a}$. Large magnitudes of $g_{0,\uparrow\downarrow}$ are required for the efficient phonon-mediated transfer of polaritons between the pseudo-spin states. The last but one column of the Table I shows the polariton threshold population $N(\text{SO})_{MP}$ required for SOs of TA- and LA’-phonons, which can be estimated using Eq. 1 with γ_{phot} replaced by γ_{MP} .

The following picture then emerges for the onset of phonoritons and related SOs. (i) The pseudo-spin energy splitting depends on the trap geometry and can change with polariton density to match the phonon energy. The strong dependence of the polariton transfer between the pseudo-spin states (mediated by phonon absorption and emission) on the splitting energy tends to equilibrate the difference in populations leading to the locking [40,49]. (ii) The pseudo-spin splitting can match the LA- or TA-phonon energy triggering the respective phonoriton mode. indeed, $g_{0,\uparrow\downarrow} \times \sqrt{N_{MP}} \approx 1 \text{ GHz} \geq \{\gamma_{MP}, \Gamma_M\}$. Such values of N_{MP} are achieved in the experiment One can see that for the TA mode, the condition for the OSC is reached for $N_{MP} \geq 8 \times 10^5$, as detailed in (*SM-II-C*). For completeness, these estimates also apply to λ -phonons. Specifically, for the LA’ λ -mode, the coupling energies are proportional to $(\lambda/\mathbf{a})^2$ and, hence, are reduced by $(1/3)^2$. However, this reduction is partially compensated by the much longer lifetime (and the higher amplitude) of the λ -phonons (cf. Fig. SM-1), which is not accounted for in Table I.

(iii) In the BEC regime, the cooperativity for the TA- and LA'-phonons can exceed unity, which leads to the self-sustained generation of the coherent phonons, i.e., the SO sidebands. From the table, one sees that TA SOs can form for $N_{MP} \leq 1000$, while the LA' SOs, observed in the more excitonic trap T2 ($X^2 = 0.2$), require much larger populations (on the order $\sim 10^5 - 10^6$ polaritons). These predictions are in qualitative agreement with the data of Fig. 2a. Table I explains the absence of f(3A)

LA sidebands in the more photonic trap T1, cf. Fig. 2a. We recall that the values in the table are calculated for a trap with the polariton excitonic fraction $X^2 = 0.5$, while for trap T1 GS $X^2 = 0.05$. Hence, LA' SOs in this trap require approximately two orders of magnitude larger N_{MP} value than the one realized in the experiment."

In the experiments related to points (i) and (ii), we could not control the population of the phonons independently of the polariton population. In the experiments pertaining to the point (iii), we precisely control the population of LA-phonons (which are generated piezoelectrically) and, hence, the magnitude of $g_{\uparrow,i}$ independently of the polariton population. Essentially, by increasing the drive of the acoustic transducer, we tune $g_{\uparrow,i}$ from low to high values and reach the condition, when the $g_{\uparrow,i} \geq \{\gamma_{MP}, \Gamma_M\}$. We observed that the linewidth reduces exactly to 1/2 of its original value, as is expected for a strong coupling. We note that since the polariton population is constant, the only mechanism that can explain the reduction is the OSC due to the increase of the $g_{\uparrow,i}$, which, as our model shows, depends on the phonon population. We included the explicit formula for the linewidth on **pages 9 (lines 314-321)**:

"The compounded mechanical and optical enhancements of the effective coupling constant relaxes the number of phonons and polaritons required for reaching the strong coupling regime. With the interaction, the simplified expression for the eigenfrequencies becomes

$$\omega_{\pm} = -j \frac{(\gamma_l + \gamma_u)}{4} \pm \sqrt{g^2 - \frac{(\gamma_l + \gamma_u)^2}{4}}$$
, where $j = \sqrt{-1}$. We assume that $\gamma_l + \gamma_u \approx 2 \times \gamma_{MP}$, then the linewidth, which is the imaginary part of above expression, can be written as:

$$\delta E = \frac{\gamma_{MP}}{2} \text{Im} \left[\sqrt{g^2 - \frac{\gamma_{MP}^2}{4}} \right]. \quad (1)$$

Hence, the the OSC condition becomes $g \geq \gamma_{MP}/2$, for which the value of δE reduces by 50%. This is due to the fact that these excitations are half mechanical and the phonon lifetime, being much larger than γ_{MP}^{-1} , virtually does not contribute to the linewidth."

Finally, we updated conclusions on **11 (lines 377-392)** to reflect the changes discussed above, which now read:

"In summary, we have demonstrated that patterned hybrid MCs with traps for polaritons and GHz phonons host novel solid-state quasiparticles – phonoritons. The latter arise from the strong coupling between two highly coherent polariton condensates and a phonon mode with energy equal to the detuning between the condensates under lateral confinement in a μm -large trap. The emergence of phonoritons has been experimentally verified by observing (i) the locking of the two polariton modes to the TA phonon energy accompanied by an additional splitting of one of the polariton modes with the coupling strength tuned by the polariton population (proportional to the optical excitation power); and (ii) the 1/2-narrowing of the condensate linewidth resulting from the enhancement of the quadratic coupling strength by the LA-phonon population. The latter demonstration takes advantage of a unique ability to precisely control the phonon amplitude using piezoelectric acoustic bulk transducers. Furthermore, the extracted magnitudes of couplings match well the ones obtained using quantitative models. The latter provide a background to understand the findings. The polariton-phonon coupling leads to enhanced optomechanical cooperativity that reaches values of 10^4 , resulting in phonon self-oscillations (phonon lasing). We also demonstrated that the phonoriton spectrum can be controlled using an external resonant laser beam. The demonstrated platform for MC phonoritons is promising for the coherent conversion between microwave and optical domains."

We believe that the changes to the text including the updates to the Figure 1 now provide a consistent explanation of the phonoriton origin in our samples.

Point 2:

”Regarding the ref. on Phonoriton, the ref cited (17) is nice. Inside, the phonoriton history is introduced going back to Keldysh in 1979, and then to some experimental works. I would say that it is probably fair at that stage to keep these old references and the rationale telling about the interest of using new systems as the one of ref. 17 and the one used in the present manuscript. This is especially true since there are no strong size constraints in nature comm.”

Response:

We agree that the description of the original concept of phonoritons as well as relevant works will improve the presentation and further highlight the relevance of our results. Following the above suggestion, we added a new paragraph on page 2 (**lines 60-78**) that overviews the works pertaining to phonoritons.

”Access to such enhanced polariton-phonon interaction regime could enable the optomechanical strong-coupling regime. Around 1982 Keldysh and Ivanov [31,32] have theoretically considered the propagation of exciton-polariton waves in a direct band gap semiconductor crystal. They showed that the interactions between the polariton waves and longitudinal acoustic phonons can lead to novel quasiparticles – the *phonoritons* – arising from the strong coupling between photons, phonons and excitons. Conceptually, due to the approximately four orders of magnitude difference between polariton and phonon energies, phonoritons require two polariton states whose energy difference matches the phonon energy. Therefore, a phonoriton can be considered as a ”dressed” polariton that resonantly emits and absorbs phonons as it propagates. The above contrasts phonoritons with conventional exciton-polaritons and phonon-polaritons, which are quasiparticles arising from the *resonant* coupling between photons and excitons, and between infrared photons and transverse optical phonons, respectively. Phonoritons eluded in-depth investigation due to the stringent experimental conditions requiring high-intensity resonant laser beams that complicate optical detection. The existence of phonoritons was suggested in early experiments, which required optical densities in the 10–10² MW/cm² range [33-35]. In contrast, Latini et al. [36] recently predicted the emergence of phonoritons in a MC with an embedded monolayer of h-BN due to the strong-coupling of two exciton-polariton resonances with phonon replicas, which is tunable by the cavity-matter coupling strength. These new phonon-exciton-photon states of matter are relevant, e.g., for novel optomechanical schemes for frequency interconversion [30], phonon and photon lasing [26], acoustic diodes [37] and the enhancement of high-temperature superconductivity [38].”

Point 3:

*”Also in the perspectives, the authors mentioned polariton based topological structures without citations. May be citing a recent review on that topic like *Optical Material Express*, 11, 1119, (2021), would be in order.”*

Response:

It is indeed an appropriate reference. Furthermore, we added the following text on **Page 11 (lines 406409)**, which suggests that phonoriton lattices could bridge the gap between purely polariton-based and phonon-based topological bands. Hence, we added the reference [56] suggested by the Reviewer as well the reference [57].

”Artificial band structures of polaritons [56] and phonons [57] have been demonstrated to host many topological effects. Excitingly, phonoriton lattices could be used to study hybrid phonon-exciton-photon topology. Furthermore, phonons can facilitate inter-trap tunneling complementary to the typically realized Josephson one [40].

Point 4:

”Page 2 fig 1.c is called instead of fig 1.b. Ref 23 and 34 are the arxiv and PRL of the same work”

Response:

We thank the Reviewer for spotting these mistakes. We carefully checked and fixed all the figure call outs and references.

FIG. 1. Microcavity phonoritons. **a** Sketch of a structured MC, which consists of a spacer embedding quantum wells (QWs) sandwiched between acousto-optic distributed Bragg reflectors (aoDBRs). The μm -wide and nm-high mesa within the spacer provides lateral confinement potential (the trap depicted by the yellow curve) for polaritons and phonons. The latter are injected optically or using a ring-shaped piezoelectric bulk acoustic wave resonator (BAWR). The phonons non-adiabatically modulate the discrete polariton energy levels (horizontal dashed yellow line) to form sidebands (dashed green lines). **b** Schematic representation of the relevant energy diagrams realized in the experiment. Full description is in the text. P_{Exc} is the optical excitation power; P_{Ths} is the condensation threshold power; GS and ES designate the trap ground and excited state, respectively; RF Off and RF On designate the conditions with the radio-frequency driving of the BAWR off and on, respectively. **c** Spatially and energy-resolved emission spectrum of trap T2 under weak non-resonant excitation. **d** Spatially integrated spectra of the same trap below the threshold (black line) and in the BEC regime (red line). **e** Spectrum of trap T1 GS as the function of the optical excitation power (P_{Exc}) normalized to the threshold power ($P_{\text{Ths}} = 60 \text{ mW}$). The levels are designated L, M and U for the lower, middle and upper one, respectively. **f** The splitting between the L and U states (ΔE_{U-L}) and between the M and U states (δ_{U-M}) as the function of $P_{\text{Exc}}/P_{\text{Ths}}$. Profiles of the map in **e** for $P_{\text{Exc}}/P_{\text{Ths}} = 1.8$ in **g** and $P_{\text{Exc}}/P_{\text{Ths}} = 2.9$ in **h**. **i** Spectrum for $P_{\text{Exc}}/P_{\text{Ths}} = 2.9$ and under piezoelectric excitation of phonons with the frequency $f_{\text{LA}}^{(3\lambda)}$

FIG. 4. **Optical coherent control of phonon sidebands.** **a** Experiment energy scheme. **b** Spectral dependence of the GS PL of trap T_1 on the control laser detuning (Δ_L) under excitation of the BAWR. The tiny white rectangles are the δ -like PL peaks induced by the resonant excitation of the sidebands (yellow arrows). The PL energy (horizontal axis) and Δ_L (vertical axis) are referenced to the zero-phonon line and scaled to the phonon energy ($hf_{LA}^{(3\lambda)}$)

The solid blue arrow indicates the position of the zero-phonon line. The diagonal line indicated by a dashed blue arrow is the Rayleigh scattering from the control laser. The inset shows a magnified region of the map plotted with the intensity in linear scale corresponding to the area defined by the dotted yellow square. **c** Integrated PL intensity of the sidebands as a function of Δ_L . **d** Reference GS spectrum in the absence of the control laser and **e-h** for selected values of Δ_L indicated by the yellow arrows. **i** An approximate energy diagram for the panel **e**. The two shifted groups of lines represent the phononiton and the control laser phonon frequency combs.

TABLE I. Calculated optomechanical coupling for the ground state (GS) and the first excited state (ES) polariton modes induced by GS 3λ phonons of TA and LA polarizations in a $4 \times 4 \mu\text{m}^2$ trap. The frequencies (ω_{BAW}) were determined for 3λ GaAs phonons and slightly underestimate the ones of the MC spacer, which also includes (Al,Ga)As layers. ΔE is the energy splitting between the pseudo-spin modes and Δ_{aa} the trap asymmetry. The phonon and polariton decay rates required for the determination of the threshold population for self-oscillation $N(\text{SO})$ and the threshold for the optomechanical strong-coupling $N(\text{OSF})$ were assumed equal to the ones of the GS $f_{LA}^{(3\lambda)}$ MP mode (see *SM-V-D* for details).

BAW mode	ω_{BAW} (GHz)	MP mode	$g_{0,\uparrow\uparrow/\downarrow\downarrow}$ (MHz)	$g_{0,\uparrow\downarrow}$ (MHz)	$N(\text{SO})_{\text{MP}} = 1/C_0$	$N(\text{OSF})_{\text{MP}}$
TA	4.6	GS	0	1.1	300	8×10^5
		ES	0	0.9	450	1.2×10^6
LA	6.5	GS	6.7	0.05^{aa}	$4.7 \times 10^5(\Delta_{aa})^{-1}$	$1011(\Delta_{aa})^{-1}$
			5.3	$0.02 \frac{\Delta_{aa}}{a}$	$7.4 \times 10^5(\Delta_{aa})^{-2}$	$1011(\Delta_{aa})^{-1}$
		ES				

REVIEWERS' COMMENTS

Reviewer #3 (Remarks to the Author):

The authors considerably improved the manuscript. I now understand much better. All my concerns have been addressed. I recommend publication in Nature Comm.

Reviewer #4 (Remarks to the Author):

I read the reviewed version of the manuscript of Kuznetsov et al.. I appreciated the answer provided by the authors to my concerns and the modifications in the manuscript. I would like to thank and congratulate the authors for the efforts they made to clarify the raised doubts. The authors have now highlighted both in the introduction and conclusions the main claim of the paper, which is the observation of phonoritons quasi-particles resulting from the strong coupling of exciton-polariton with acoustic phonons confined in micro-cavities traps. I think the manuscript now provides a detailed description of resonance phenomena which are of great interest for the community, together with inspiring experimental observations. I recommend publication of the manuscript without further review and I congratulate the authors for their job.

Response to Reviewers.

Reviewer 3

The Reviewer recommends to publish without further changes with the following statement:

"The authors considerably improved the manuscript. I now understand much better. All my concerns have been addressed. I recommend publication in Nature Comm."

Reviewer 4

The Reviewer recommends to publish without further review as stated in their response:

"I read the reviewed version of the manuscript of Kuznetsov et al.. I appreciated the answer provided by the authors to my concerns and the modifications in the manuscript. I would like to thank and congratulate the authors for the efforts they made to clarify the raised doubts. The authors have now highlighted both in the introduction and conclusions the main claim of the paper, which is the observation of phonoritons quasi-particles resulting from the strong coupling of exciton-polariton with acoustic phonons confined in micro-cavities traps. I think the manuscript now provides a detailed description of resonance phenomena which are of great interest for the community, together with inspiring experimental observations. I recommend publication of the manuscript without further review and I congratulate the authors for their job."